# Haploidy in somatic cells is induced by mature oocytes in mice

Yeonmi Lee[1,2,8], Aysha Trout[3,8], Nuria Marti-Gutierrez[4,8], Seoon Kang [1,5], Philip Xie[3], Aleksei Mikhalchenko[4], Bitnara Kim[1], Jiwan Choi[1,5], Seongjun So[1,5], Jongsuk Han[1], Jing Xu[4,6,7], Amy Koski [4], Hong Ma[4], Junchul David Yoon [4], Crystal Van Dyken [4], Hayley Darby[4], Dan Liang[4], Ying Li[4], Rebecca Tippner-Hedges[4], Fuhua Xu[4,7], Paula Amato[4,7], Gianpiero D. Palermo [3✉], Shoukhrat Mitalipov [4✉] & Eunju Kang [1,2✉]

Haploidy is naturally observed in gametes; however, attempts of experimentally inducing haploidy in somatic cells have not been successful. Here, we demonstrate that the replacement of meiotic spindles in mature metaphases II (MII) arrested oocytes with nuclei of somatic cells in the G0/G1 stage of cell cycle results in the formation of *de novo* spindles consisting of somatic homologous chromosomes comprising of single chromatids. Fertilization of such oocytes with sperm triggers the extrusion of one set of homologous chromosomes into the pseudo-polar body (PPB), resulting in a zygote with haploid somatic and sperm pronuclei (PN). Upon culture, 18% of somatic-sperm zygotes reach the blastocyst stage, and 16% of them possess heterozygous diploid genomes consisting of somatic haploid and sperm homologs across all chromosomes. We also generate embryonic stem cells and live offspring from somatic-sperm embryos. Our finding may offer an alternative strategy for generating oocytes carrying somatic genomes.

[1] Department of Biomedical Science, College of Life Science, CHA University, Seongnam, Gyeonggi 13488, South Korea. [2] Center for Embryo and Stem Cell Research, CHA Advanced Research Institute, CHA University, Seongnam, Gyeonggi 13488, South Korea. [3] The Ronald O. Perelman and Claudia Cohen Center for Reproductive Medicine, Weill Cornell Medicine, New York, NY 10021, USA. [4] Center for Embryonic Cell and Gene Therapy, Oregon Health and Science University, Portland, OR 97239, USA. [5] Department of Medical Science, Asan Medical Institute of Convergence Science and Technology (AMIST), University of Ulsan College of Medicine, Asan Medical Center, Seoul 05505, South Korea. [6] Division of Reproductive and Developmental Sciences, Oregon National Primate Research Center, Oregon Health and Science University, Portland, OR 97006, USA. [7] Department of Obstetrics and Gynecology, Oregon Health and Science University, Portland, OR 97239, USA. [8] These authors contributed equally: Yeonmi Lee, Aysha Trout, Nuria Marti-Gutierrez. ✉email: gdpalerm@med.cornell.edu; mitalipo@ohsu.edu; ekang@cha.ac.kr

Haploidization, or the reduction of ploidy to a single set of chromosomes, ensues naturally during gametogenesis, where the segregation of homologous chromosomes occurs during meiosis I (MI)[1]. For generating meiosis II (MII) oocytes, the reductional MI division is followed by an equational MII division, similar to mitosis, where the sister chromatids are segregated to opposite spindle poles, eliminating one set into the second polar body.

Several studies attempted to induce somatic haploidy. However, the diploid somatic cells that transferred into enucleated oocytes to achieve the haploid nucleus resulted in limited development of the preimplantation embryos[2]. In mice, the somatic cells such as cumulus or fibroblasts were transferred to immature germinal vesicle (GV) or mature MII oocytes to induce haploid chromosomes, but the reconstructed chromosomes exhibited abnormalities in the separation and alignment processes and embryo development was not observed[3–6]. In humans, the cumulus cells were injected into the MII ooplasm after chromosomes of the oocyte were removed. The reconstructed oocytes were fertilized with a spermatozoon resulting in the polar body extrusion, which showed the segregation of homologous chromosomes in several chromosomes. However, no further development was observed[7].

Here, we revisited the haploidization of diploid somatic chromosomes using the somatic cell nuclear transfer (SCNT) technique, in which the somatic cell nucleus was transferred into enucleated metaphase MII-arrested oocyte. We examined meiotic spindles in SCNT oocytes that were produced by transplanting a $G_0/G_1$ somatic cell depending on resting time after SCNT and confirmed the chromosome segregation after in vitro fertilization (NT-IVF). We also improved the rates of development of somatic haploid (SH) embryos with various combinations of chemicals or protein. Finally, the SH embryos established embryonic stem cells (ESCs) and produced offspring. This study could provide a new strategy to generate oocytes carrying somatic genomes.

## Results

### The formation of the meiotic spindle–chromosomal complex from $G_0/G_1$ somatic chromosomes.

We postulated that the somatic chromosomes with pseudo-meiotic spindles originating from the diploid $G_0/G_1$ nucleus (2n) of the somatic cell are composed of two single chromatids (2n/2c). These chromatids can be segregated by fertilization and haploid somatic pronucleus (1n/1c) could be produced in the zygote (Fig. 1a).

We imaged SCNT oocytes generated from somatic cells under a noninvasive polarized microscope. The spindles were not observed within 30 min after SCNT. The newly formed spindles first became visible 1 h 30 min after SCNT and were clearly organized 2 h after it (Fig. 1b and Supplementary Movie 1). In addition, α-tubulin, a protein required for chromosome segregation during cell division, was stained at 1, 1.5, 2, and 3 h after SCNT. The microtubules were not detected until 1 h and exhibited a prometaphase-like arrangement at 1.5 h (Fig. 1c). The comparable spindle–chromosomal complex to that in intact MII oocytes was observed at 2 h. The anaphase-like microtubule arrangement was detected at 3 h. Moreover, the SCNT oocytes, as well as control MI and MII oocytes, were fixed and labeled with α-tubulin, a protein of microtubules, and kinetochore, a multiprotein complex that assembles on centromeric DNA and constitutes the main attachment interface between chromosomes and microtubules[8]. As expected, the kinetochores in the control MI oocytes appeared as punctate, parallel signals (red) at the equatorial region of the spindle (Fig. 1d). Chromosomes were stretched by centromere–kinetochore pairs toward opposite spindle poles. By contrast, MII spindles carried one row of

centromeres with microtubules connected on both sides of each centromere. The chromosomal arrangement pattern in some SCNT spindles was similar to those observed in MI oocytes.

We next examined the ability of the SCNT spindles to segregate chromosomes into the pseudo polar body (PPB) and SH zygote via NT-IVF. The PPB extrusion rate was evaluated in SH zygotes, which were rested for 30 min as well as 1, 1.5, 2, and 3 h after SCNT. The SCNT oocytes that were rested for 2 h exhibited a significantly higher PPB extrusion rate than that of other groups (Fig. 1e, Supplementary Table 1, and Supplementary Data 1).

Spindles in the SH zygote disappeared gradually, a PPB was extruded, and two pronuclei (PN) were formed (Fig. 1f). Staining confirmed that the PPB and both PN contained DNA. The timing of these events and the morphology of the SH zygote were indistinguishable from those of the control.

### Fasudil, retinoic acid, and RS-1 promote the segregation of homologous chromosomes.

The SCNT technique has been performed as described previously[9]. Briefly, a hemagglutinating virus of Japan envelope was applied to fuse the donor somatic cells with the enucleated oocytes. After resting time for 30 min to 1 h, the reconstructed oocytes were activated with strontium and HDAC inhibitors such as Trichostatin A or Scriptaid. Based on this conventional method, we modified the protocol for NT-IVF, which was extended resting time for 2 h. Additionally, the caffeine was treated before and during SCNT micromanipulation to prevent premature oocyte activation and to prompt spindle reformation in SCNT oocytes[10,11]. The normal SH zygote morphology was two PN and one PPB (2PN/1PPB), while irregular SH zygotes were 2PN/0PPB, 1PN/1PPB, 3PN/0PPB, and 1PN/0PPB (Fig. 2a).

To improve the somatic haploidy, fasudil (ROCK, rhoassociated protein kinase, inhibitor), retinoic acid (RA), and RAD51-Stimulatory Compound 1 (RS-1) were treated in SCNT oocytes or SH embryos. The first, the fasudil was treated during IVF. Since ROCK supports spindle assembly in mature oocytes[12], fasudil might assist the spindle decomposition during fertilization and enhance the PPB extrusion through the regulation of microtubule polarity[13]. The rate of 2PN or 2PN/1PPB formation was calculated based on the number of fertilized embryos. Approximately 75% of the SH zygotes formed 2PN (Supplementary Fig. 1a). The yield of SH zygotes with the proper morphology, 2PN/1PPB, in the fasudil-treated group ($n = 58/184$, 32%) was significantly higher than that of such SH zygotes in the untreated group ($n = 24/109$, 22%; $P < 0.05$), whereas blastocyst rates were similar between the groups (Fig. 2b, Supplementary Table 1, and Supplementary Data 1).

RA initiates the entrance of the prophase of meiosis I during oogenesis[14,15]. Because we proposed that the premature chromosomes from the $G_0/G_1$ somatic cell could be similar to the prophase of meiosis I of the oocyte, we tested several incubation times in SCNT oocytes for 30 min, 1 h, and 2 h (Supplementary Fig. 1b and Supplementary Table 1). Treatment with RA for 30 min produced the highest number of 2PN/1PPB SH zygotes compared with the other groups (52%, 46/89 for 30 min group vs. both 24%, 20/83 for 1 h and 17/70 for 2 h groups, respectively, $P < 0.05$). The 2PN/1PPB rate in the RA-treated group was significantly higher than that in the untreated group (51%, $n = 104/202$ vs. 32%, $n = 58/184$; Fig. 2c, Supplementary Table 1 and Supplementary Data 1). We then observed the spindle formation in SCNT oocytes using a noninvasive imaging system to examine the effect of RA (Fig. 2d). In the RA-treated group, 59% ($n = 65/110$) SCNT oocytes developed spindles, which was significantly higher than the rate in the untreated group (44%, $n = 46/104$; $P < 0.05$). By contrast, once the novel spindle was formed, the rates of 2PN/1PPB were comparable between the

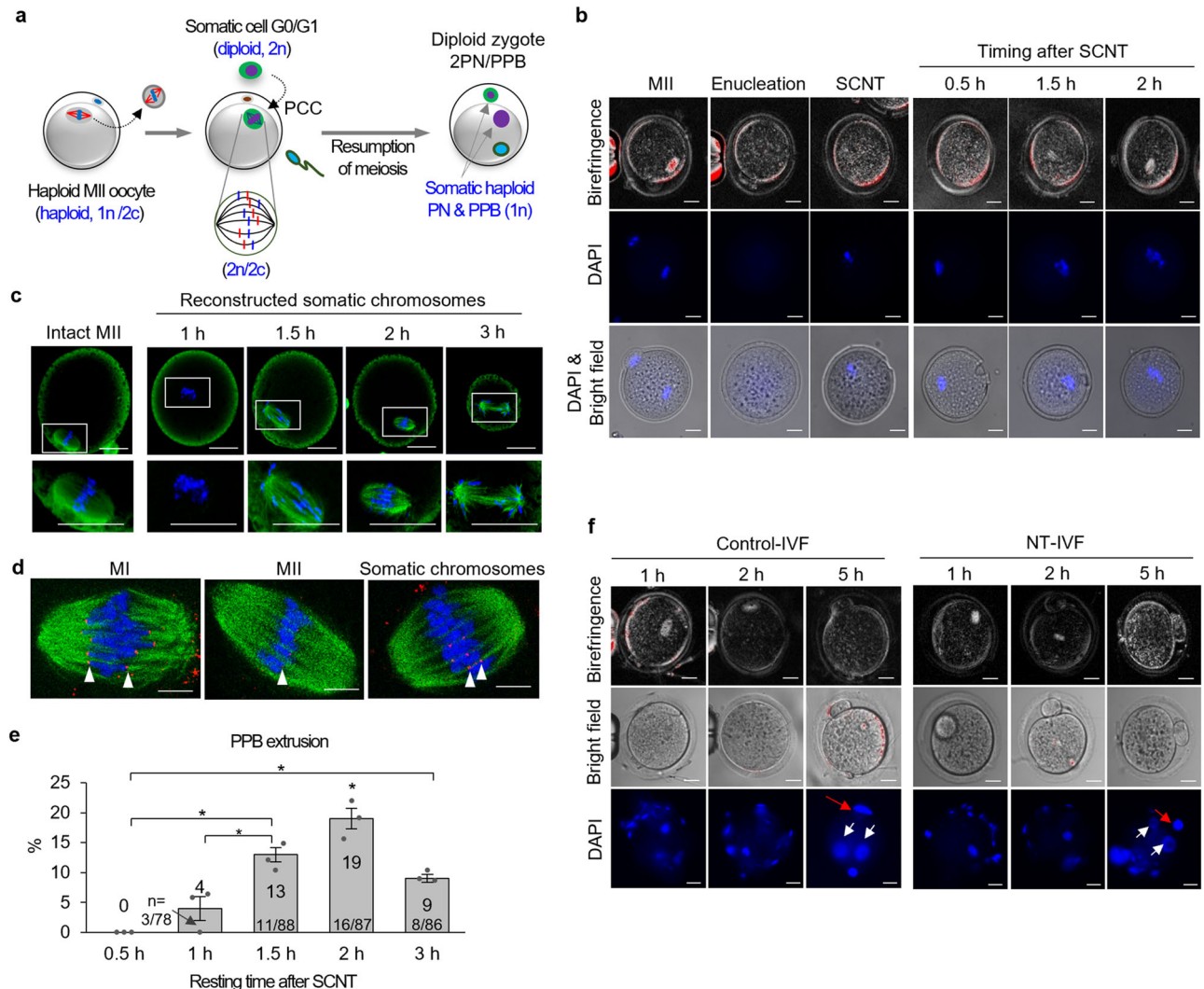

**Fig. 1 Phenotypical observation of somatic nucleus haploidization by mature oocytes. a** Schematic of diploid somatic nucleus haploidization by mature ooplasm. SCNT oocyte with pseudo-meiotic spindle derived from diploid $G_0/G_1$ somatic nucleus (2n) could be composed of two single chromatids (2n/2c), and chromatids could be segregated (1n) at fertilization. Half of the segregated chromatids (1n/1c) could be extruded to the pseudo-polar body (PPB), whereas the rest (1n/1c) remained in a zygote with a sperm nucleus. **b** Spindle reformation in SCNT oocytes. The spindle was cleared 2 h after SCNT. Scale bars, 20 μm. **c** Spindle-chromosomal complexes in intact MII and SCNT oocytes. Metaphase-like spindle-chromosomal complex was observed at 2 h (blue, DNA; green, α-tubulin). Scale bars, 25 μm. **d** Spindle–chromosome complexes in intact MI, MII, and SCNT oocyte. SCNT spindles showed a similar chromosomal arrangement to MI oocyte (blue, DNA; green, α-tubulin; red, kinetochore). Scale bars, 5 μm. **e** PPB extrusion depending on resting time after SCNT. 2 h resting after SCNT resulted in significantly higher PPB extrusion than that in other groups. mean ± s.e.m. (*$P < 0.05$, by ANOVA with Tukey analysis) among the groups. $n$ means the number of 2PN/1PPB embryos/the number of fertilized embryos. Three technical replications for each group. **f** Spindle and nuclear changes after fertilization in intact MII and SCNT oocytes. SCNT oocytes showed PPB extrusion (red arrows) and 2 PN formation (white arrows), Scale bars, 20 μm.

treatment groups (92%, 60/65 in RA vs. 89%, 41/46 in RA free; 2PN/1PPB zygotes/spindle reconstructed oocytes; Fig. 2d and Supplementary Data 1).

Finally, we evaluated the effect of RS-1, which enhances the expression of the homologous recombinases, such as Rad51 and Dmc1[16]. The RS-1 could support the alignment of homologous chromosomes of the somatic nucleus in SCNT oocytes thus increasing PPB extrusion. To determine the optimum treatment timing, we applied RS-1 during the resting time after SCNT, during IVF, and overnight after IVF (Supplementary Fig. 1c and Supplementary Table 1), resulting that the treatment during IVF resulted in a significantly higher rate of 2PN/1PPB than that in other groups. 2PN/1PPB rate with RS-1 treatment during IVF was significantly higher compared to the non-treatment (67%, $n = 148/221$ vs. 51%, $n = 104/202$, $P < 0.05$;

Fig. 2e, Supplementary Table 1, and Supplementary Data 1). However, the rates of blastocysts were comparable. Based on the results of these treatments, we established an advanced (a)NT-IVF protocol; RA was treated for 30 min after SCNT, rested SCNT oocytes for 2 h before IVF, and scriptaid, fasudil, and RS-1 were treated during IVF and the overnight culture (Fig. 2f).

These treatments could promote the spindle reconstruction and formation of normal SH zygotes. The SH zygotes had a normal morphological development up to the blastocyst stage (Fig. 2g and Supplementary Movie 1). The incorporation of these treatments significantly increased 2PN/1PPB (67%, $n = 148/221$, vs. 17%, $n = 31/184$; 2PN/1PPB/fertilized) and blastocyst (50%, $n = 27/54$ vs. 29%, $n = 4/14$; blastocysts/morula) formation rates compared with non-treated ($P < 0.05$; Fig. 2h and Supplementary Table 1). When the blastocyst rate was calculated from MII

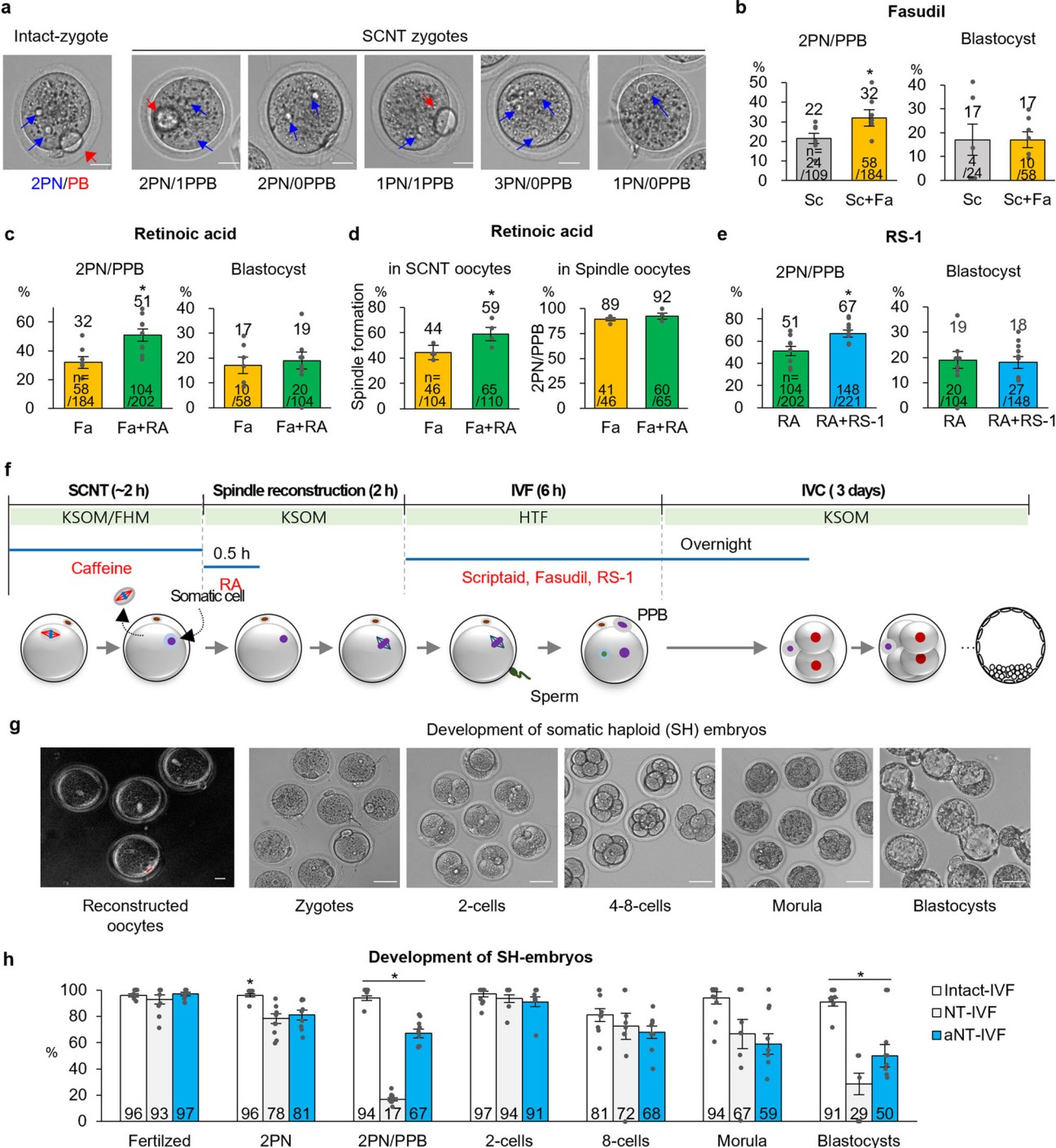

oocytes, 12% oocytes ($n = 27/229$) could develop blastocysts with the aNT-IVF protocol, while only 2% ($n = 4/198$) with the non-treated NT-IVF one (Supplementary Table 2). The blastocyst rates were 91% ($n = 138/152$; blastocysts/morula) and 69% ($n = 11/16$) in intact IVF and regular SCNT, respectively. As we expected, intact IVF showed a significantly higher blastocyst rate than that of aNT-IVF (Fig. 2h, Supplementary Table 1, and Supplementary Data 1). However, the rate of blastocysts tended to be higher in regular SCNT (69%) compared to aNT-IVF (50%).

**Reciprocal segregation of somatic homologous chromosomes.**
Although morphological examinations of SCNT oocytes and SH zygotes are informative and conclusive, evaluations of somatic

chromosomal haploidization can only be determined by genetic analysis of chromosomal content. We initially conducted whole-exome sequencing (WES) analyses of FVB/N (FVB) and C57BL/6 N (B6) mouse strains and determined homozygous single-nucleotide polymorphisms (SNPs) that differentiate each strain (Supplementary Fig. 2a and Supplementary Data 2). A catalog representing unique high-confidence SNPs could distinguish FVB from B6 chromosomes. We designed primers amplifying one region (1100 to 1500 bp) with the highest SNP frequency in each of 19 autosomal chromosomes and sequenced those regions using the MiSeq platform for FVB or B6 genotyping (Supplementary Fig. 2b, c).

To determine the segregation pattern of somatic chromosomes, we generated SCNT oocytes by transferring FVB/B6 (a cross

**Fig. 2 Optimization of the somatic cell haploidization protocol. a** Various morphologies of somatic haploid (SH) zygotes. A normal SH zygote (2PN/1PPB), which was similar to the intact control zygote (2PN/PB), was observed. Other morphologies of SH zygotes were also produced. Blue arrows: PN; red arrows: the second PB and PPB in the intact control zygotes and SH zygotes, respectively. Scale bar, 20 μm. **b** Improved 2PN/1PPB formation with Fasudil (Fa) treatment during IVF (*$P < 0.05$, by Independent-group $t$ test). Sc, scriptaid. **c** Enhanced 2PN/1PPB formation with retinoic acid (RA) treatment during the resting time (*$P < 0.05$, by Independent-group $t$ test). **d**, Increased spindle formation in SCNT oocytes with RA treatment. The spindle formation rate was significantly increased with RA treatment compared with the RA-free condition (*$P < 0.05$, by Independent-group $t$ test). However, once the spindle was reformed, the rates of 2PN/1PPB became comparable. **e** Improved 2PN/1PPB formation with RS-1 treatment during IVF (*$P < 0.05$, by Independent-group $t$ test). **f** Advanced schematic protocol for somatic haploidization. Caffeine (250 μg/ml) was supplemented before and during SCNT. RA (300 ng/ml) was treated for 30 min after SCNT. After RA treatment, the SCNT oocytes were rested for 1.5 h before IVF. Sc (80 ng/ml), Fa (3 μg/ml), and RS-1 (4 μg/ml) were added to the medium during IVF and the overnight culture. **g** SCNT oocytes with reconstructed spindles and the development of SH embryos. The SCNT oocytes with reconstructed spindles were fertilized, and the SH zygotes showed normal development up to the blastocyst. Scale bar, 20 μm. **h** Improved development of preimplantation embryos with the advanced protocol (aNT-IVF). aNT-IVF significantly increased the rates of 2PN/1PPB and blastocysts compared with NT-IVF. (*$P < 0.05$, by ANOVA with Tukey analysis) among the groups. $n$ in the 2PN/1PPB graphs and the blastocyst graphs of **b**, **c**, and **e**; the number of 2PN/1PPB embryos/the number of fertilized embryos and the number of blastocysts/the number of 2PN/1PPB embryos, respectively. Six–10 technical replications for **b**, **c**, **e**, and **h**, and four technical replications for **d**. mean ± s.e.m.

between female FVB and male B6) or B6/FVB (the opposite combination: a cross between female B6 and male FVB) somatic cells. The SCNT oocytes were fertilized with B6 sperm (Fig. 3a). A total of 15 SH zygotes, 10 from FVB/B6 and 5 from B6/FVB, with normal 2PN/1PPB appearance, were cultured to the 2-cell stage. PPB and blastomeres were separated to perform whole-genome amplification (WGA) for the assessment of the segregation of the 19 autosomes using MiSeq (Supplementary Data 3, sheet 1). The X chromosome was analyzed by Sanger sequencing (Supplementary Fig. 3a). The segregation of somatic FVB (red) and B6 (blue) homologous chromosomes from the somatic cells were observed in most PPBs and SH embryos (Fig. 3b). Initially, we hypothesized that FVB SNPs could only be detected in either PPB or SH embryo because the somatic donor was heterozygous. However, the exome sequencing showed the recombined homozygous SNPs in heterozygous somatic donors (Supplementary Fig. 3b and Supplementary Data 2), which could make to detect the FVB SNPs in both PPB and SH embryo. Therefore, if FVB SNPs were detected in SH embryos and their corresponding PPB, we also considered the proper segregation of somatic chromosomes. We first checked the zygosity of chromosomes in each PPB, resulting in that 10–20 chromosomes were homozygous, which could be haploidy segregated from somatic genomes (Fig. 3c). Among them, either FVB or B6 chromosomes were identified randomly in each chromosome. Next, the number of properly segregated chromosomes into PPB and SH embryos was analyzed. In total, 9–20 homologous chromosomes were properly segregated between SH embryos and their corresponding PPBs (Fig. 3d). Some chromosomes in PPBs showed heterozygosity (74%) or were not amplified (26%), suggesting that these homologous chromosomes were not separated and extruded to PPBs or remained in embryos (Fig. 3e and Supplementary Data 1). In the SH embryos, 66% and 68% of haploid chromosomes in the FVB/B6 and B6/FVB combinations were originated from the FVB strain respectively, suggesting that the somatic genome remaining in SH embryos after haploidization was more species-specific rather than maternally or paternally biased (Fig. 3f and Supplementary Data 1). We also analyzed the segregation for each chromosome in 15 SH embryos. Chromosome 1 was segregated properly in all 15 embryos, whereas the other 19 chromosomes were separated in 8–14 embryos (Supplementary Fig. 3c).

In addition, SCNT oocytes transferred with FVB/B6 somatic cells were chemically activated without sperm. Ten PPBs were isolated and analyzed for homozygosity in all chromosomes (Supplementary Fig. 3d and Supplementary Data 3, sheet 2). Five PPBs were detected FVB or B6 homozygous genotype (GT) in all chromosomes, suggesting that chemical activation

could also promote the segregation of somatic homologous chromosomes.

Next, we selected four pairs of 2-cell embryos and their corresponding PPBs, two from FVB/B6 (SH embryos 3 and 9) and two from B6/FVB (SH embryos 11 and 14) somatic cells, and performed WES (Supplementary Data 2). In the FVB/B6 combination, the copy numbers of chromosomes, analyzed by exome data, showed that 13 (in SH embryo 3) and 11 (in SH embryo 9) homologous chromosomes were properly segregated (Fig. 4a, b). The remaining chromosomes were nondisjunct and either extracted to PPBs (green boxes in Supplementary Fig. 4a, b) or remained in SH embryos (blue boxes in Supplementary Fig. 4a, b). In the properly segregated homologous chromosomes, the homologous chromosomes were separated as completely FVB or B6 between the PPB and the embryo in the whole-exome area (Fig. 4c, d and Supplementary Fig. 4a, b). Of those, 6 and 7 chromosomes, respectively, were of the FVB genome in embryos.

By contrast, in both embryos from the B6/FVB combination, the copy numbers of chromosomes were properly segregated in all 20 chromosomes (Fig. 4e, f). However, unlike in the FVB/B6 combination, both the PPB and the embryo contained the FVB genome (Fig. 4g, h and Supplementary Fig. 5a, b). Much of the shared FVB genome between the PPB and the embryo showed homozygosity in somatic donor cells (Supplementary Fig. 5c, d), indicating that the shared FVB SNPs were derived from homozygous SNPs of somatic cells and the homozygous FVB SNPs were segregated into the PPB and the embryo. However, some shared SNPs between the PPB and embryo were not carried in somatic cells as homozygous status, because (1) every single somatic cell transferred into an ooplast could harbor different SNPs, and (2) the pooled somatic cell cannot represent all homozygous SNPs of every single cell.

These results suggest that two single homologous chromosomes (2n/2c) in the SCNT oocytes were segregated randomly after fertilization, producing truly haploid cells from somatic cells in zygotes.

**Contribution of somatic genomes in all chromosomes of SH embryos**. To determine the contribution of somatic chromosomes to SH embryos, we used adult fibroblasts derived from female homozygous FVB strains to generate SCNT oocytes. SCNT oocytes were fertilized with B6 sperm, and embryos at the 2-cell or blastocyst stage were examined for all 20 chromosome GTs (Fig. 5a).

A total of 19 2-cell stage embryos and 19 blastocysts were screened in chromosome 2 and sex chromosomes (Supplementary Fig. 6a, b). The FVB (somatic origin) and B6 (sperm origin) genomes were detected in nine 2-cell embryos (47%) and 11

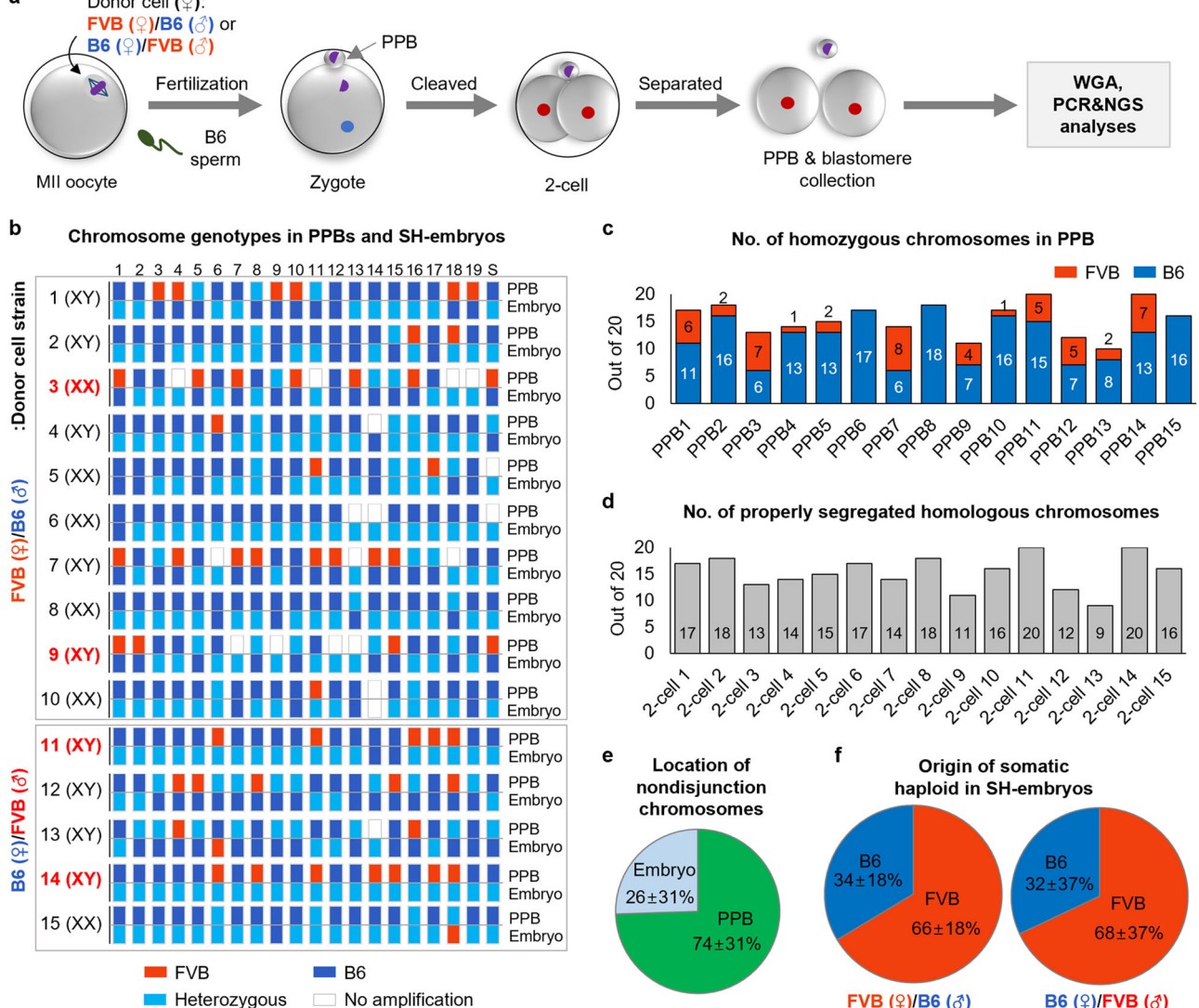

**Fig. 3 Chromosome segregation of somatic homologous chromosomes. a** Schematic procedure to determine the segregation pattern of somatic chromosomes. FVB/B6 or B6/FVB fibroblasts were transferred into enucleated oocytes and fertilized with B6 sperm. The 2PN/1PPB 2-cell embryos were separated into SH embryos and PPB, and artificial whole-genome amplification (WGA) was performed for genotyping. **b** Chromosome genotypes in PPBs and SH embryos using MiSeq. Proper segregation patterns of somatic homologous chromosomes were displayed in most PPBs and SH embryos. Red bars, FVB; blue, B6; light blue, heterozygosity with FVB and B6; white, the absence of PCR amplicons; S above the last bars, sex chromosome. **c** Number of homozygous chromosomes in PPBs. 10–20 chromosomes showed haploid, which could be haploidy segregated from somatic genomes. The genotype between FVB and B6 was random in each chromosome. **d** Number of properly segregated homologous chromosomes. Properly segregated chromosomes were analyzed by considering the homozygosity of donor cells as revealed by exome sequencing. Nine to twenty homologous chromosomes were segregated reciprocally between SH embryos and corresponding PPBs. **e** Location of nondisjunction chromosomes. 74% of nonseparated homologous chromosomes were located in PPBs. mean ± s.d. **f** Origin of somatic haploid in SH embryos. In all, 66–68% of haploid chromosomes were originated from FVB. Mean ± s.d.

blastocysts (58%) in chromosome 2 (Supplementary Fig. 6c). Sex determination revealed that 55% ($n = 21/38$) of the SH embryos were male, indicating that these embryos inherited their X chromosome from the somatic cell genome (Fig. 5b and Supplementary Fig. 6d).

The 20 FVB-positive embryos in chromosome 2 were genotyped for all autosomes using MiSeq (Supplementary Data 3, sheet 3 and 4), and the X chromosome of the female embryos was analyzed by Sanger sequencing (Supplementary Fig. 7a). The results showed that three 2-cell embryos ($n = 3/19$, 16%) and six blastocysts ($n = 6/19$, 32%) were FVB/B6 heterozygous across all 20 chromosomes, suggesting the presence of SH embryos in all 20 chromosomes with somatic haploidy (Fig. 5c, d and

Supplementary Fig. 7b, c). All three 2-cell embryos and five blastocysts were male. The remaining 11 embryos included B6 or FVB homozygosity in a few chromosomes, which could indicate partial haploidization in those embryos or amplification errors (Supplementary Data 3, sheets 3 and 4). The increase in SH in blastocysts could be due to the arrest of many aneuploid SH 2-cell embryos before reaching the blastocyst stage.

To confirm the somatic origin and euploidy in whole chromosomes, we performed WES for selected male blastocysts (BL6, BL10, BL12, and BL14; Supplementary Data 2). The FVB genomes were detected in the whole chromosomes in 3 blastocysts (BL6, BL10, and BL14), but not in chromosomes 1, 8, 9, 18, and 19 of blastocyst 12 (Fig. 5e and Supplementary

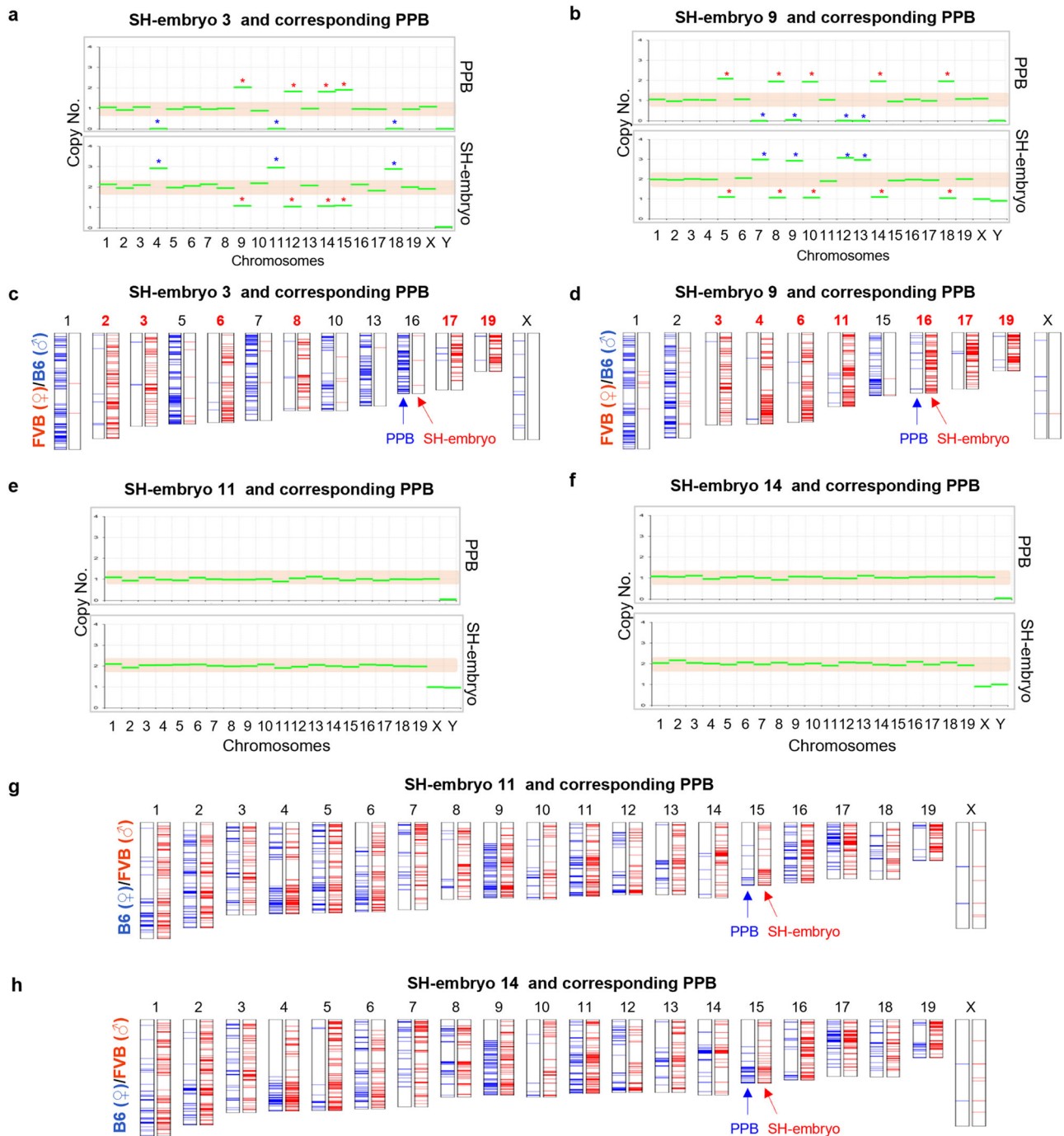

**Fig. 4 Somatic chromosome segregation to SH embryos and corresponding PPBs. a, b** Copy number variation (CNV) profiles of SH embryos 3 and 9 and their corresponding PPBs generated by FVB/B6 somatic donor. In total, 13 and 11 homologous chromosomes were properly segregated in SH embryos 3 and 9, respectively. The other chromosomes were nondisjunct and extracted to PPBs (red asterisk) or remained in SH embryos (blue asterisk). Relative CNV was interpreted by comparison with the control IVF embryo, second PB, and C57BL/6 mouse tissue as a control. **c, d** Chromosome map of SH-embryo 3 and 9 and their corresponding PPB. Thirteen (in SH-embryo 3) and 11 (in SH-embryo 9) chromosomes were segregated reciprocally. **e, f** CNV profiles of SH embryos 11 and 14 and their corresponding PPBs generated by B6/FVB somatic donor. Copy numbers of chromosomes displayed proper segregation of somatic homologous chromosomes in all 20 chromosomes of both embryos. **g, h** Chromosome map of SH-embryo 11 and 14 and their corresponding PPB. All chromosomes were segregated reciprocally in both embryos.

Fig. 7d). The normal euploidy copy number of all chromosomes was detected in blastocysts 6, 10, and 14, suggesting complete somatic haploid in all chromosomes (Fig. 5f). As expected, in blastocyst 12, chromosomes 1, 8, 9, 18, and 19 were haploid with only the sperm genome, whereas chromosome 6 was triploid (Supplementary Fig. 7e).

**Global gene expression in SH-ESCs.** Because the analysis of embryos had limitations due to WGA or technical errors, ESCs from SH blastocysts (SH-ESCs) were established using B6/FVB somatic cells to validate the results found in the embryos. We generated several SH-ESC lines, and the efficiency of ESC derivation was 7% (4 ESCs/55 NT-IVF BL), significantly lower than

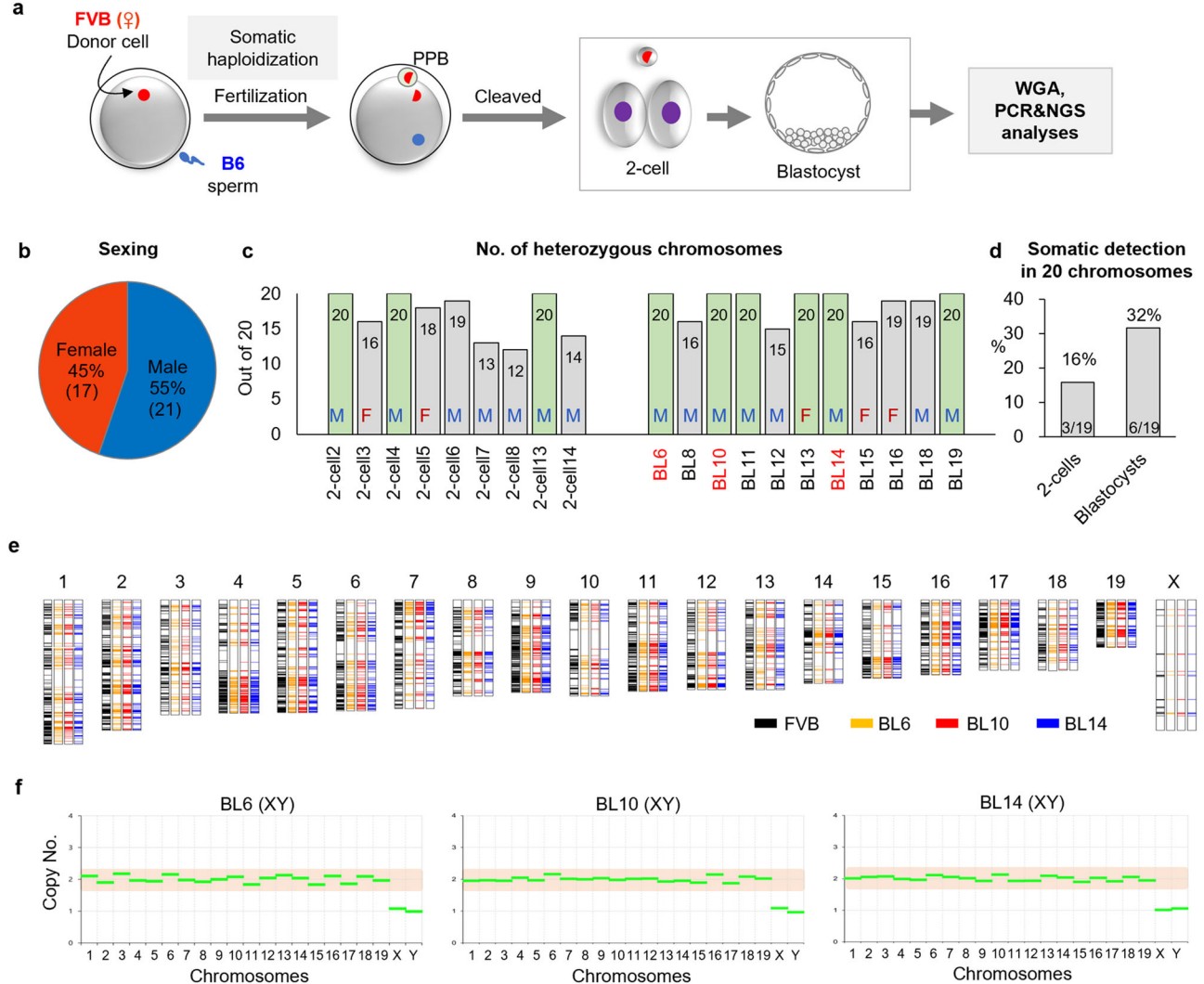

**Fig. 5 Contribution of the somatic origin in SH embryos. a** Schematic illustrating the contribution of somatic chromosomes to SH embryos. The adult fibroblasts derived from homozygous FVB mice were used for somatic cell nuclear transfer. SCNT oocytes were fertilized with B6 sperm. Artificial whole-genome amplification was performed on 2-cell embryos and blastocysts and analyzed all 20 chromosome genotypes. **b** The sex ratio of the SH embryos. Of all the embryos, 22 (55%) were male and 17 (45%) were female. **c** The number of heterozygous chromosomes. Among the blastocysts showing heterozygosity in all 20 chromosomes, WES was performed for BL6, 10, and 14 (red font). M and F indicate male and female, respectively. **d** The frequency of somatic detection in 20 chromosomes. Three 2-cell embryos ($n = 3/19$, 16%) and 6 blastocysts ($n = 6/19$, 32%) harbored the somatic origin in all 20 chromosomes. **e** Chromosome map of the male SH blastocysts. FVB genomes were confirmed by WES in blastocysts 6, 10, and 14 (somatic origin) in all 20 chromosomes. **f** CNV profile of the SH blastocysts by exome data. Euploidy was shown in blastocysts 6, 10, and 14 in whole chromosomes.

that of ESC from IVF embryos (75%, 13 ESCs/23 IVF BL) (Supplementary Fig. 8a). These SH-ESCs were demonstrated a typical morphology and a normal diploid karyotype (Fig. 6a–d and Supplementary Figs. 8b and 13). The single male cell line, with the biopsied sample of the same blastocyst, was further investigated (Supplementary Fig. 8c). The results showed that 3 or 4 chromosomes were heterozygous in the SH embryo derivatives in regions analyzed using MiSeq (Fig. 6e). The remaining chromosomes harbored the B6 GT. The typical 2n copy number for all 20 chromosomes was confirmed in biopsied blastocysts and SH-ESCs (Fig. 6f). The GTs by exome sequencing revealed that the FVB genome was similarly displayed between the blastocysts and SH-ESCs (Fig. 6g and Supplementary Data 2). Chromosomes 2, 5, and 11 originated mainly from the FVB, whereas chromosome 13 did not harbor any FVB SNPs in the SH-ESCs.

We examined the global gene expression patterns of SH-ESCs using RNA-seq in comparison with intact ESCs. A total of 22,014 genes were expressed in at least one of the intact ESCs or SH-

ESCs. Only 200 genes were determined with a significant $P$ value ($P < 0.05$), and the clustering of these 200 genes resulted in a separation between the intact ESC and SH-ESC lines (Fig. 7a). Among them, seven genes with significant adjusted p-values were considered to be differentially expressed (false discovery rate, FDR < 0.05) in SH-ESCs (Fig. 7b and Supplementary Fig. 8d). Six genes (*GM29100*, *Lef1os1*, *8030451A03Rik*, *Sncg*, *Gm41724*, and *Gm21992*) were expressed at a significantly higher level, whereas one gene (*Antxrl*) was expressed at a significantly lower level in SH-ESCs (Fig. 7c). The phenotype or detailed in vivo function of all these genes, except *Sncg*, has not been reported yet (Supplementary Fig. 8e). The *Sncg* is known to be related to neurodegenerative disease based on its overexpression[17], which could interrupt the full development of SH embryos in vivo.

Because the regulation of imprinting genes is important for embryonic and fetal growth or reprogramming to a pluripotent state, a total of 105 gene expressions were investigated in SH-ESCs[18]. We focused on several imprinting genes, such as *H19*,

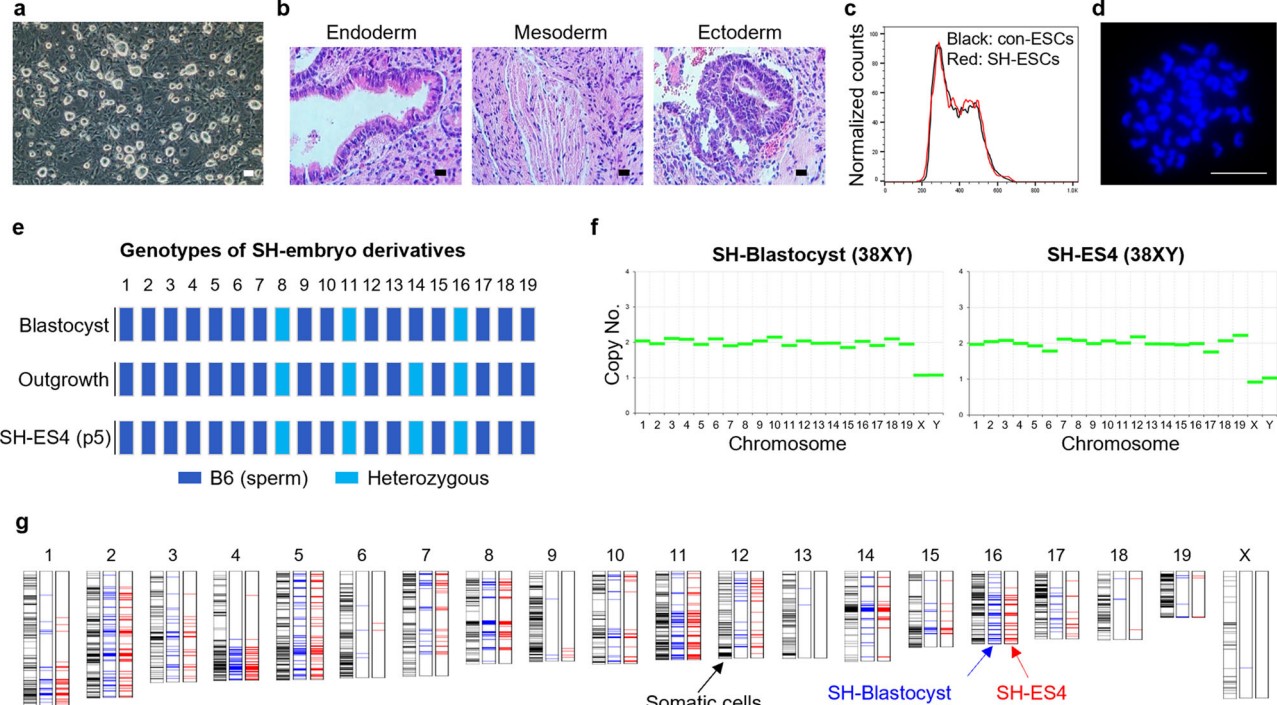

**Fig. 6 Somatic origin in established SH-ESCs. a** The morphology of the SH-ESCs. SH-ESCs (SH-ES4) derived from SH blastocysts showed the typical morphology of mouse ESCs. Scale bar, 100 μm. **b** Three germ layer formations of the SH-ESCs by teratoma assay. Scale bars, 100 μm. **c** Diploid configuration of the SH-ESCs by cell cycle analysis. The histogram refers to the cell cycle profile of the SH-ESCs resulting in a 2n nuclear configuration. **d** Representative image of the diploid SH-ESCs chromosome spread of all 40 chromosomes. Scale bars, 10 μm. **e** Genotype of SH embryo derivatives by MiSeq. Outgrowth refers to an inner cell mass of outgrowth in the SH blastocyst-plated dish. Blue bars: B6 genotype; light blue: heterozygous status with FVB and B6. Three or four chromosomes were heterozygous in SH embryo derivatives and the remaining chromosomes harbored the B6 genotype. **f** Copy number variation profile of the SH-blastocysts and SH-ES4 with exome data. Diploid copy numbers were displayed in all 20 chromosomes. Relative CNV was interpreted by comparison with the controls, in vitro fertilization embryo, second polar body, and C57BL/6 mouse tissue. **g** Chromosome map of SH-blastocysts and SH-ES4. The distribution of FVB SNPs was similar between the blastocysts and the SH-ES4.

*Igf2r*, and *Grb10* as paternally imprinted genes, and *Igf2* and *Snrpn* as maternally imprinted genes (Supplementary Fig. 8f). The results suggested that these genes showed no significant difference in SH-ESCs compared with intact ESCs (*P* > 0.05). The remaining imprinting genes (100 genes) also displayed no significant difference between SH-ESCs and intact ESCs. Based on these results, we concluded that the gene expressions of SH-ESCs were comparable to those of intact ESCs.

**SH-embryos are able to produce live offspring**. To evaluate the full-term development of SH embryos, we generated SH blastocysts using somatic cells from various strains, including FVB fetal and adult fibroblasts, B6/FVB fetal fibroblasts, as well as B6D2F1/Crl (BDF1, a cross between C57BL/6NCrl female x DBA/2NCrl male) and FVB cumulus cells (Fig. 8a and Supplementary Fig. 9a). The rates of 2PN/1PPB were comparable among donor strains, in approximately 65%, but were lower than the rates in the unmanipulated IVF controls (94%). BDF1 donors yielded higher blastocyst rates among donor strains (30% vs. 11–18%).

Only SH blastocysts with a good morphology were transferred into recipients (Supplementary Fig. 9b). The quality of SH blastocysts was poor compared with that in the unmanipulated IVF controls (33% vs. 96%; Supplementary Fig. 9c). Only 8.6% 2PN/1PPB zygotes were developed to blastocysts evaluated as being of good quality (*n* = 18/232), compared with 65% (*n* = 132/206) in the controls (*P* < 0.05; Supplementary Fig. 9d). In addition, 9 SH blastocysts underwent trophectoderm biopsy and were subjected to WGA to determine embryo ploidy with

copy number variation (CNV) analysis using Illumina. Of these, three were normal euploid (33%, *n* = 3/9), of which two were female and one was male (Supplementary Fig. 9e, f).

A total of 95 blastocysts were generated using FVB somatic cells and B6 sperm were transferred into nine recipients, resulting in no pregnancies (Fig. 8b and Supplementary Fig. 9g). The somatic cells that originated from the B6/FVB hybrid mouse also failed to result in pregnancy (118 blastocysts to 10 recipients). The B6/FVB hybrid somatic cells with sperm from BDF1 hybrid mice improved the pregnancy rate. A total of 121 blastocysts were transferred into 35 recipients, resulting in 3 pregnancies. The implanted embryos from two recipients were lost, but the uterus from the third recipient was collected on day 7 after embryo transfer. The three implanted spots (embryos) were observed in the collected uterus, and their DNA was extracted (Supplementary Fig. 9h). Array comparative genomic hybridization was performed to detect euploidy in the three SH embryos. All three embryos had a diploid genome, and genomic alterations were not detected (Fig. 8c). One of these embryos (SH embryo 1) was selected to analyze chromosomal CNV using WES data, with the results confirming the diploidy (Fig. 8d). The FVB origin in SH-implanted embryo 1 was analyzed using WES data (Supplementary Data 2). We conducted WES of DBA/2 and determined FVB-specific SNPs compared with DBA/2. All 20 chromosomes contained FVB-specific SNPs (Fig. 8e). FVB genetic bias occurred in the SH-implanted embryos, similar to the case of SH-ESCs.

Finally, 81 SH blastocysts combined with BDF1 somatic cells and BDF1 sperm were transferred into 27 recipients, resulting in one pregnancy and the delivery of 3 female pups (Fig. 8f). The

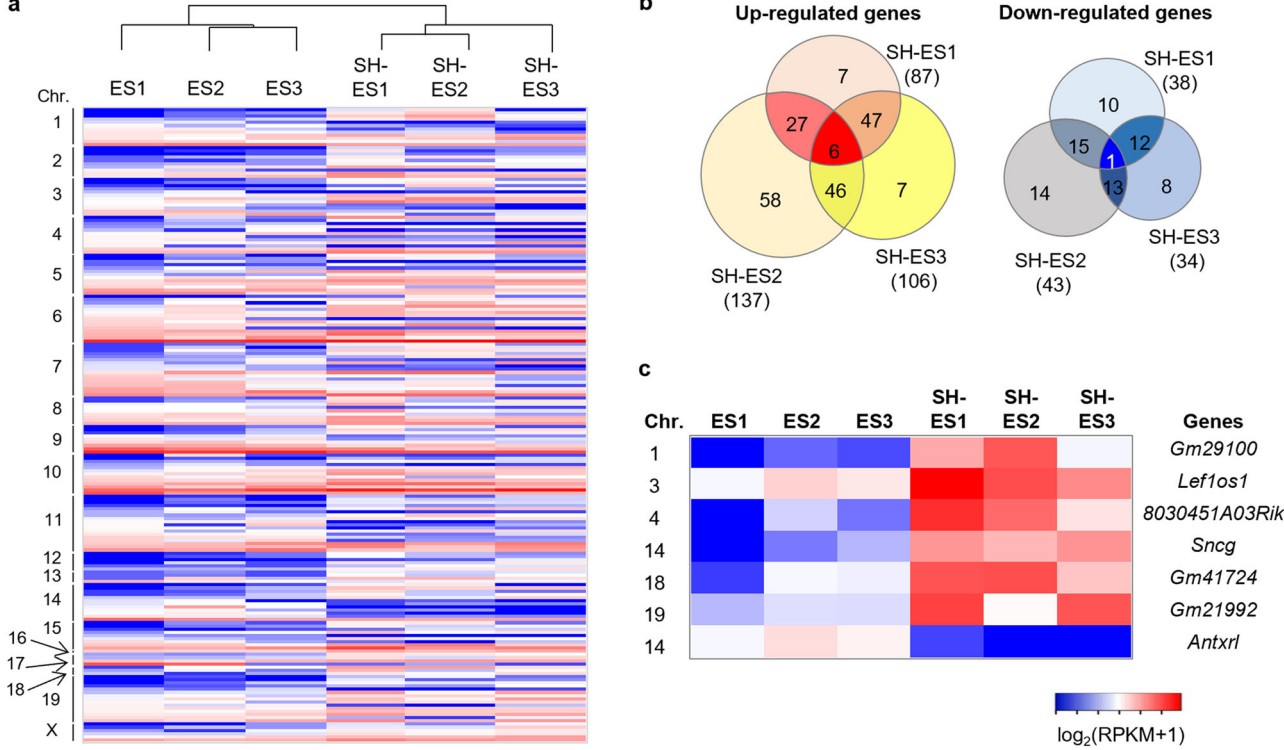

**Fig. 7 Global gene expression in SH-ESCs. a** Heat map displaying 200 genes with significant difference (*P* < 0.05) between intact ESCs and SH-ESCs. The clustering of gene expression with a significant *P* value resulted in a separation between intact ESC and SH-ESC lines. **b** Venn diagram showing the number of upregulated or downregulated genes in SH-ESCs compared to intact ESCs. **c** Heat map displaying seven differentially expressed genes between intact ESCs and SH-ESCs (false discovery rate, FDR < 0.05). Six genes were upregulated, whereas, one gene was downregulated in SH-ESCs.

average bodyweight of the pups was 1.1 g, which was significantly lower than that of the intracytoplasmic sperm injection (ICSI) controls (1.66 ± 0.17 g) (Fig. 8g, Supplementary Fig. 9i and Supplementary Data 1). All three pups survived and grew into adulthood (Fig. 8f). Furthermore, all SH mice were mated with BDF1 males and produced three healthy first-generation litters, which had birth weights similar to those of ICSI controls (Fig. 8g, Supplementary Fig. 9j, and Supplementary Data 1). All first-generation litters of the SH mice survived and grew into adulthood (Fig. 8f).

## Discussion
This study successfully generated haploid chromosomes from somatic cells using mature oocytes (Fig. 9). Meiotic spindles were formed after the transfer of somatic cells into enucleated MII oocytes. After fertilization of reconstructed oocytes, SH zygotes formed 2PN and extracted PPBs. As the initial rate of PPBs was low (19%), we added several chemicals such as fasudil (ROCK inhibitor), RA (miosis initiator), and RS-1, to improve the rate of PPB extraction. Indeed, the rate increased to 67%, albeit still lower than the rate in unmanipulated IVF controls (95%). In particular, RS-1 is known as a Rad51 enhancer, which is a recombinase responsible for homologous recombination. Rad51 is approximately 45% structurally similar to Dmc1, which is responsible for homologous centromere coupling and homologous chromosome pairing[16]. Therefore, RS-1 likely supports the alignment and segregation of somatic homologous chromosomes. All SH embryos in this study showed evidence of somatic haploidization, such that the haploid chromosomes were segregated reciprocally between the PN and the PPB in several homologous chromosomes.

The somatic chromosome segregation and nuclear remodeling/reprogramming could be crucial for the successful generation of

offspring harboring haploid genomes derived from somatic cells. After the diploid somatic nucleus could be segregated to haploid in ooplasm, then the proper nuclear remodeling/reprogramming is required, resulting in the full development of SH embryo and the generation of SH offspring. Normal nuclear reprogramming to produce cloning animals has been proven in multiple species, even the efficiency was low[19].

However, there are debates for the success of chromosome segregation of somatic nucleus[2,20]. The first attempt of somatic haploidization was performed using the mature oocytes (MII) in human[7]. Reconstructed oocytes were fertilized, resulting in the extrusion of PPB. These PPBs were confirmed a single fluorescence signal by fluorescence in-situ hybridization in five chromosomes, which could indicate the segregation of homologous chromosomes. Other investigators tried somatic haploidization using immature oocytes (GV) in humans and mice[3]. After transplantation of somatic cells into the GV ooplasm, the extrusion of the first polar body was observed in both humans and mice. However, the rate of polar body extrusion was under 1% and abortive metaphase plates were observed in mice[4]. Later, somatic haploidy was tried using MII oocytes in mice by Tateno and colleagues[6,20]. Reconstructed oocytes were chemically activated and most oocytes, that extruded PPB, failed to leave a haploid number of chromosomes. None of the reconstructed chromosomes showed metaphase-like-array before PPB extrusion. Based on the results, they claimed there was no opportunity for the pairing of somatic homologous chromosomes in MII oocytes and non-random chromosome segregation could not have happened in MII oocytes[20]. Our study also used MII oocytes and demonstrated that reconstructed oocytes showed a metaphase-like spindle–chromosomal complex and the PPB was confirmed as haploidy by copy number analysis. Proper segregation of homologous chromosomes was observed

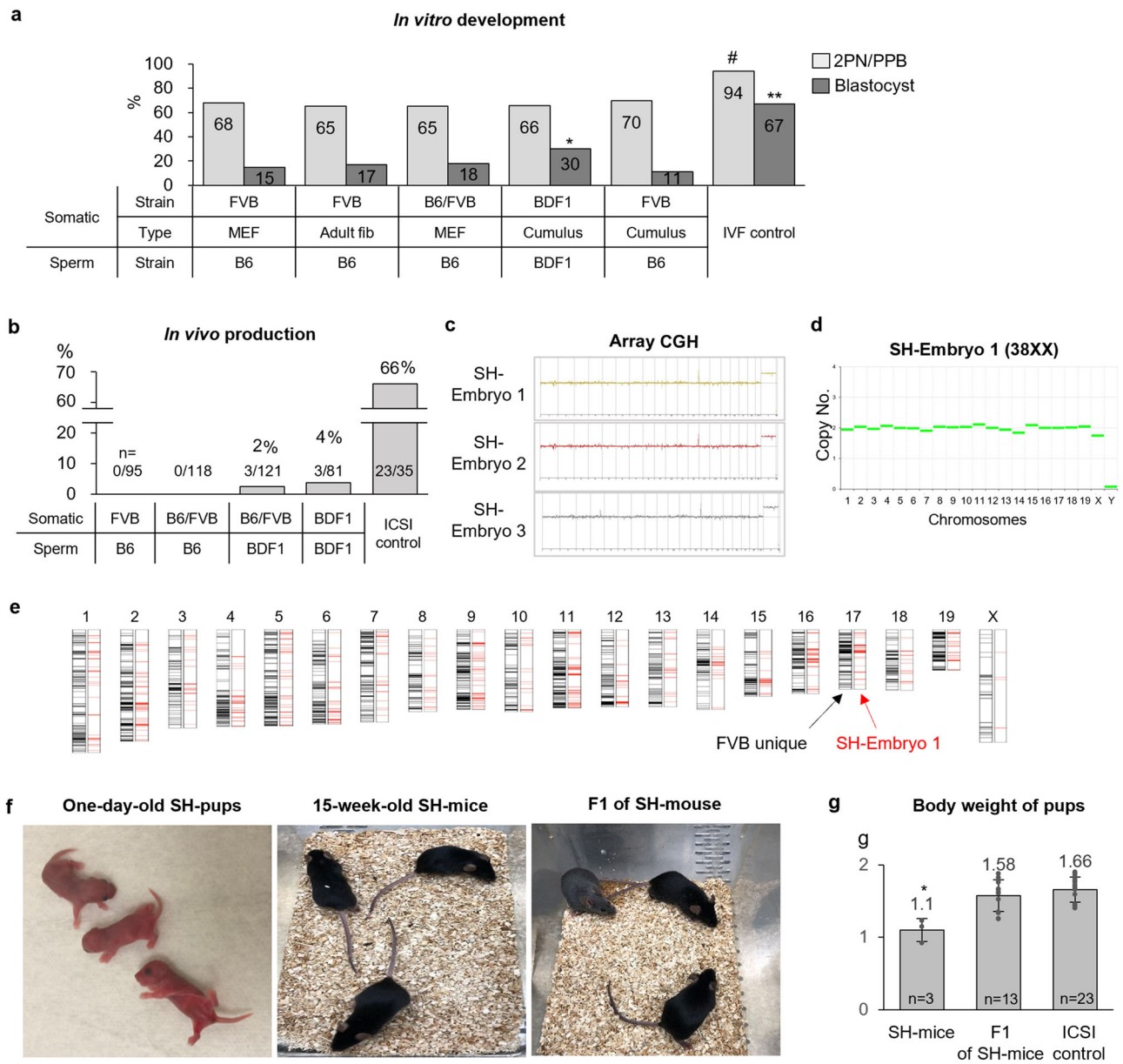

**Fig. 8 Somatic haploid embryos are able to produce live offspring. a** In vitro development depends on the somatic cell and sperm. The rates of 2PN/1PPB were comparable among donor strains, type, and sperm strain, whereas BDF1 donor cells showed higher blastocyst rates than other donor strains. *,#*P* < 0.05; ***P* < 0.01 between the two groups by Fisher's exact test. **b** In vivo production from SH embryos with different strain combinations of somatic cell and sperm. *n* indicates the number of in vivo production/number of transplanted blastocysts. **c** Diploidy of the SH-implanted embryos. SH-implanted embryos had a diploid genome in all chromosomes by array comparative genomic hybridization analysis, and the results for the SH-implanted embryos were analyzed by comparison with DBA/2 mouse tissues as a control. **d** CNV profile of SH-implanted embryo 1 by exome sequencing. CNV using exome sequencing data showed diploidy in SH-implanted embryo 1. Relative CNV was interpreted by comparison with the controls, in vitro fertilization-embryo, second PB, and C57BL/6 mouse tissue. **e** Chromosome map of SH-implanted embryo 1. FVB unique indicates specific FVB SNPs of somatic cells against those of DBA/2. All 20 chromosomes contained FVB-unique SNPs. **f** SH mice on day 1 (left) and 15 weeks (middle) after birth and the first generation (F1) of SH mice (right). **g** The body weight of SH mice and F1 of SH mice after birth. The body weights of the SH mice were significantly lower than those of SH-F1 and intact intracytoplasmic sperm injection control pups. Mean ± s.d. *n* means the number of mice for each group. **P* < 0.05, by ANOVA with Tukey analysis.

in all analyzed SH zygotes, the number of the properly segregated chromosome in each SH embryos was 9 to 20. We suggested that our modified SCNT protocol could assist the proper chromosome segregation.

Further, Tateno and colleagues suggested that the chance of retaining chromosomes of just one parental origin, known as semi-cloning, is rare, which could be $<1 \times 2^{-20}$ for 20 chromosome pairs in mice, therefore, semi-cloning did not have any

advantage for the cloning with somatic haplodization[20]. Therefore, we did not concern about the semi-cloning during haplodization because we assumed that it was not necessary to transmit only just one parental origin after somatic haplodization. We demonstrated that the contribution of somatic chromosomes in SH embryo was random between maternal or paternal alleles in the somatic genome. Instead of the odds of semi-cloning, we focused on the proper segregation of somatic homologous

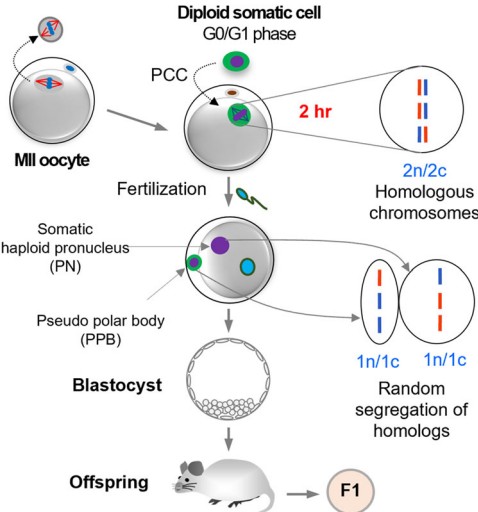

**Fig. 9 Schematic graphic for induction of haploidy in somatic cells by mature oocytes.** The somatic cell was transferred into enucleated MII oocyte and the meiotic spindle was reconstructed. Fertilization prompts homologous segregation and formation of zygote consisting of haploid somatic and sperm pronuclei. The embryos can develop to live offspring.

chromosomes to PPB and embryo, resulting in an average of 76% of the homologous chromosome being properly segregated in 15 SH embryos (45–100% of range in each embryo).

The fundamental question that this study sought to answer is how somatic diploid chromosomes can segregate to haploid chromosomes in MII ooplasts. A previous report discovered female meiosis in humans[21]. Canonically, during prophase I of meiosis, homologous chromosomes are recognized and paired[22]. The first meiotic division (MI) separates pairs of homologous chromosomes, and the second division (MII) separates sister chromatids. However, in certain cases, homologous chromosomes are separated during MII, not MI, a process known as reverse meiosis. In mouse oocytes, if sister chromatids were separated at MI for some reason, non-sister chromatids or homologous chromosomes could be separated at the MII stage[1,21]. The molecular mechanism behind this remains unknown. Ottolini et al. (2015) suggested that either the oocytes were physically attached by unresolved recombination of joint molecules or other threads or used segregation mechanisms that are not related to the physical attachment between chromosomes[21]. In conclusion, MII oocytes are capable of separating homologous chromosomes into embryos and PBs reciprocally.

The SH embryos developed into blastocysts and produced live offspring, but with very low efficiency. One explanation for this low development efficiency is that chromosome loss due to chromosome segregation failure could lead to SH embryo developmental arrest. The second explanation could be the limitations inherent in SCNT, such as inappropriate nuclear reprogramming and abnormal placental development, which could hinder further in utero development[23]. Previous studies found that various epigenetic abnormalities, such as DNA methylation, histone modifications, and genomic imprinting, induce a low live birth rate from SCNT embryos[19]. The third explanation is the mismatch of the reprogramming cycle between sperm and somatic haploidy genome. Initially, we expected that the reprogramming of NT-IVF might be better than regular SCNT because one nucleus of the reconstructed embryo was originated from a germ cell (sperm). However, the blastocyst development from the morula was ~50% in NT-IVF, which was tended to be lower than regular SCNT (69%). Therefore, we supposed that if the embryo

nuclei were originated from the same cell type, such as oocyte and sperm nucleus in intact IVF embryos or somatic cell nuclei in regular SCNT embryos, the reprogramming and embryo development could be more effective. Even somatic haploidization was successful in NT-IVF embryos, sperm and somatic genome harbored different nuclear statuses for reprogramming, which could make the development arrest.

The generation of SH offspring was available by only the combination of BDF1 somatic cells and BDF1 sperm in this study. Initially, we expected that the combination of B6/FVB somatic cells and BDF1 sperm could produce the SH offspring because hybrid species are efficient to produce offspring experimentally than inbred species[24]. However, this combination failed. Another combination, BDF1 somatic cells, and BDF1 sperm were successful, suggesting that the species was crucial for producing full-term births experimentally. Unfortunately, BDF1 was not used consistently throughout the genetic analyses, therefore, it is hard to confirm the haploidization in SH pups, which is a limitation of our study.

Approximately 10%–15% of couples suffer from infertility[25,26]. Moreover, 3.5% of women with infertility who undergo assisted reproductive technology (ART) have no oocytes upon retrieval[27]. To date, none of the available techniques can be applied to these women for getting their babies[28]. Combined with the generation of artificial oocytes from pluripotent stem cells[29,30], our technology can apply next-generation ART, including mitochondrial replacement therapy in which somatic cells from patients with infertility or mitochondrial disease could be introduced into the cytoplasm of enucleated artificial oocytes.

## Methods

**Animals**. B6D2F1/Crl (C57BL/6NCrl female × DBA/2NCrl male) female mice (8–9 weeks old, Charles River) were used as oocyte recipients. FVB/N female and male (8–9 weeks old, Taconic Biosciences), C57BL/6N female and male (8–9 weeks old, Charles River), and BDF1/Crl female (8–9 weeks old) were used for the generation of somatic donor cells for SCNT. ICR female mice (10–12 weeks old, Charles River) were used as recipients for embryo transfer. All animal maintenance and experimental procedures were performed following the guidelines of the Asan Medical Center, Oregon Health & Science University, and Weill Cornell Medicine. The animal protocols were reviewed and approved by the Asan Medical Center Institute, Small Lab Animal Unit at Oregon Health & Science University, and Weill Cornell Medicine Institutional Animal Care and Use Committee (IACUC) before any experiments were performed. The mice were housed under a 12-h shift of light/dark cycle under pathogen-free conditions with free access to water and food.

**Isolation and culture for donor cells**. Female MEFs generated from FVB/B6 (FVB/N females x C57BL/6 males), B6/FVB (C57BL/6 females x FVB/N males), or homozygous FVB/N mice and adult fibroblasts isolated from homozygous FVB/N female mice were used as the donor cells for SCNT. MEFs were established from embryos at 13.5 dpc and the embryos' heads and organs were removed before cell isolation. Adult fibroblasts were isolated from ear skin tissue at 8 weeks old. Tissue was dissociated with 0.1% collagenase IV, incubated for 30 min, and diluted with an equal volume of F12/DMEM media (Gibco10099141) supplemented with 10% FBS (Gibco). The cells were cultured in F12/DMEM with 10% FBS, 100 units/ml penicillin (Hyclone), 100 µg/ml streptomycin (Hyclone), 100 µM β-mercaptoethanol (Sigma), and 100 µM nonessential amino acid (Gibco) under 5% $CO_2$ at 37 °C in a humidified incubator. Cumulus cells were collected from BDF1 or FVB/N mice and kept in KSOM (Merck) until nuclear transfer.

**MII oocyte collection**. Female BDF1 mice were super-ovulated with 5 international units (IU) pregnant mare's serum gonadotropin (PMSG) or 0.2 ml of ready-to-use PMSG/inhibin (CARD Hyperova) and 5 or 7.5 IU human chorionic gonadotropin (hCG). MII oocytes were collected from the excised oviducts 13–16 h after hCG injection and kept in KSOM (Merck) drops supplemented with 250 µg/ml caffeine (Sigma) until SCNT.

**Somatic cell nuclear transfer**. For SCNT, oocytes were placed into a manipulation droplet of FHM or M2 medium containing 5 µg/ml cytochalasin B (Sigma) and 250 µg/ml caffeine (Sigma) in a glass-bottom dish (FluoroDish, WPI). The dish was mounted on the stage of an inverted microscope (Nikon) equipped with a stage warmer, micromanipulator (Narishige), Oosight® Imaging System (Hamilton Thorne), and a XyClone® laser objective (Hamilton Thorne). The spindle in the

oocyte was placed close at the 2 to 4 o'clock position using a holding pipette. The zona pellucida next to the spindle was drilled with a laser, an enucleation pipette was inserted into the cytoplasm, and a small amount of cytoplasm containing the spindle was aspirated into the pipette. Alternatively, an oocyte was held at 9 o'clock by a holding pipette to perform enucleation and was oriented to position the meiotic spindle at 12 o'clock. A laser (Hamilton Thorne) was used to breach the zona pellucida at 12 o'clock. Next, a slight pressure was applied by an injection pipette (Piezo Drill Tip, Eppendorf, Enfield) at the 3-o'clock position, creating a protrusion containing a meiotic spindle through the breach. The injection pipette was then used to divide the protruding karyoplast from the remaining ooplasm inside the zona pellucida. Next, an HVJ-E extract (GenomONE™)-treated fibroblast or cumulus cell was aspirated into a micropipette and transferred into the enucleated oocyte[9]. The SCNT oocytes were treated with various chemicals for 2 h before IVF.

**Sperm preparation and IVF of the SCNT oocytes**. The epididymis was collected from a male mouse, transferred to Human Tubal Fluid (HTF) media (Merck), and cut to make the sperm swim out. After the sperm was released, the epididymal tissues were removed from the HTF drop. The sperm were incubated under 5% $CO_2$ at 37 °C in a humidified incubator for 30 min. The SCNT oocytes were transferred to a new HTF media drop. Sperm swimming at the edge of the HTF drop was collected and transferred to the SCNT oocytes contained in the HTF drop. The SCNT oocytes were treated with various chemicals during IVF, and IVF was performed for 6 h under 5% $CO_2$ at 37 °C in a humidified incubator.

**Spermatozoa collection and ICSI of SCNT oocytes**. The cauda epididymis of BDF1 male mice was surgically excised and the spermatozoa were released into HTF medium by microdissection. The spermatozoa were incubated under 5% $CO_2$ at 37 °C for at least 3 h before use for insemination. The spermatozoa were resuspended in HTF medium to achieve a concentration of 3 million/ml for piezo-ICSI.

For piezo-ICSI, an injection pipette with a 25° tip angle, 6 μm inner diameter, 6 mm long was back-loaded with Fluorinert (Sigma) and attached to a micromanipulator equipped with a piezo actuator (PMM-150FU Piezo Impact Drive, Prime Tech). Excessive air and a small amount of Fluorinert were expelled into a PVP drop in the manipulation dish and the pipette was then washed with PVP. Mouse sperm heads were mechanically separated from their tails and aspirated into the injection pipette. While holding the reconstructed oocyte at 9 o'clock by a holding pipette, the injection pipette was inserted through the breach created in the zona during the enucleation step at the 3-o'clock position. The injection pipette was passed 80% through the oocyte forming an invagination, and a piezo pulse was used to breach the membrane and deposit the sperm head into the ooplasm. The pipette was removed while slightly aspirating to close the breach in the oolemma. Several unmanipulated MII oocytes were ICSI inseminated to generate control embryos. Piezo-ICSI was performed similarly except that the zona was breached using the piezo pulse rather than a laser.

**SH-embryo culture and recoding of development**. SH-embryos with two pronuclei and pseudo-polar bodies were transferred to KSOM media (Merck) supplemented with various chemicals and cultured overnight under 5% $CO_2$ at 37 °C in a humidified incubator for 12 h. The next morning, the embryos were moved to KSOM medium and cultured under 5% $CO_2$, 5% $O_2$ at 37 °C in a humidified incubator.

The ICSI oocytes were placed in a time-lapse incubator (EmbryoScope, Vitrolife). Full preimplantation development was recorded; every embryo development event was annotated for up to 96 h with images taken every 10 min. Timing of the embryo development hallmarks was compared between the control and experimental groups.

**Establishment and culture of mouse ECSs (mESCs)**. Denuded blastocysts were placed onto mitomycin C (Sigma)-treated mouse embryonic fibroblast (MEF) feeder layers in mESC derivation medium: KODMEM (Gibco) containing 20% KOSR (Gibco), 1 mM L-glutamine (Gibco), 100 units/ml penicillin (Hyclone), 100 μg/ml streptomycin (Hyclone), 100 μM β-mercaptoethanol (Sigma), 100 μM nonessential amino acid, 10 μM fasudil (Adooq), 0.5 μM PD0325901 (PeproTech), 3 μM CHIR99021 (PeproTech), and 1000 units/ml LIF (Stemgent) under 5% $CO_2$ at 37 °C in a humidified incubator[9,31]. The cell outgrowth was dissociated using trypsin-EDTA and seeded on the new MEF feeder layer. Established mESCs were passaged every 3–4 days using trypsin-EDTA for further experiments.

**Immunocytochemistry**. Oocytes were fixed using 2% formaldehyde for 20 min and permeabilized with 0.1% Triton X-100 for 20 min at room temperature (RT). Oocytes were blocked with 0.3% BSA for 10 min at RT and incubated with primary antibodies against centromere (Antibodies Incorporated, 1:30) for 1 h at RT. Oocytes were washed three times and incubated with secondary goat anti-human IgG Alexa Fluor 568 (Invitrogen, 1:200) and α-tubulin rabbit mAb Alexa Fluor 488 (Cell Signaling Technology, 1:100) for 1 h at 37 °C. The oocytes were washed 3 times and mounted in VECTASHIELD with DAPI (Vector Laboratories).

**Whole-genome amplification**. Embryos, such as blastomeres, PPBs, and second PBs were transferred into 0.2 mL PCR tubes and placed in a freezer at −20 °C. WGA was performed using a REPLI-g Single Cell Kit (Qiagen)[32] according to the manufacturer's specifications. The amplified DNA was used for further analysis.

**Miseq**. Miseq was performed as previously described[33] with minor modifications. The DNA was amplified at 1100–1500 bp in 19 autosomes. The primers for amplification are shown in Supplementary Fig. 2c. The PCR reaction was performed under the following conditions: one cycle at 98 °C for 30 s, then 35 cycles at 98 °C for 10 s, 58 °C for 10 s, 72 °C for 2 min, followed by one cycle at 72 °C for 5 min with Platinum™ SuperFi™ PCR Master Mix (Invitrogen). The concentration of the PCR products was measured by the Qubit 2.0 Fluorometer (Invitrogen). Library preparation was performed using a Nextera XT DNA sample preparation kit (Illumina) following the manufacturer's instructions. Sequencing was performed on the Illumina MiSeq platform using Miseq reagent Kit v2 (300 cycles, Illumina), and the data were analyzed by the NextGENe software (Softgenetics). Briefly, sequence reads ranging from 250 to 500 bps were quality filtered and processed using the NextGENe software and an algorithm similar to BLAT. The sequence error correction feature (condensation) was performed to reduce false-positive variants and produce sample consensus sequences and variant calls for each sample. After the quality check, the raw FASTQ reads were filtered and converted to the FASTA format. Filtered reads were aligned to the C57BL/6 genome sequence reference (GCF_000000055.19, GRCm38.p6) followed by variant calling.

**Sex determination using PCR**. The sex of embryos and ESCs were determined by PCR assay using the following primers: mSexF: CTGAAGCTTTTGGCTTTGAG and mSexR: CCACTGCCAAATTCTTTGG. The PCR reaction was performed under the following conditions: one cycle at 95 °C for 5 min, then 35 cycles at 95 °C for 20 s, 55 °C for 20 s, 68 °C for 20 s, and followed by one cycle at 68 °C for 3 min.

**Teratoma formation**. Teratoma formation experiments were performed by injecting SH-ESCs into the femoral region of 7-week-old NOD-SCID Gamma mice (Gembioscience) using a 1 ml syringe with an 18-gauge needle. Five weeks after cell injection, the mice were euthanized, and the teratomas were isolated, sectioned, and histologically characterized for the presence of representative tissues of all three germ layers using hematoxylin and eosin staining.

**Cell cycle analysis**. Dissociated cells were fixed with 70% ethanol at 4 °C for 15 min. The cells were washed twice and stained with 50 μg/ml propidium iodide (PI, Sigma) solution containing 100 μg/ml RNase A at 4 °C for 40 min. The cell cycle was measured using a FACS Calibur (BD Biosciences), and the data were analyzed by FlowJo v10.5.3 (FlowJo LLC). In all, 10,000 single-cell events were adjusted in the pulse width versus pulse area plot. The purity of single-cell events was >90% excluding debris and death cell populations. The plot was applied to the PI histogram plot. All samples were applied the same gating strategy.

**Karyotyping**. KaryoMAX Colcemide (Gibco) at a final concentration of 150 ng/ml was applied to the cultured cells for 1.5 h at 37 °C. The treated cells were then detached by trypsin/EDTA. After hypotonic treatment with 0.075 M KCL for 30 min, the cells were fixed with methanol: acetic acid (3:1 v/v). The fixed cells were dropped onto a slide, and the slides were mounted in Prolong Diamond Antifade Mountant with DAPI (Invitrogen). Data acquisition was performed on a fluoroscopic microscope (AxioObserver Z1; Carl Zeiss, Oberkochen, Germany), and image acquisition was performed with Zen 2 (Zeiss).

**Exome sequencing and data analysis**. DNA was extracted from the samples using Gentra Puregene tissue kit (Qiagen), and qualified DNA proceeded to library preparation using the SureSelect Mouse exome library kit (Agilent Technologies). Qualified DNA was randomly fragmented, and the adapter-ligated fragments were amplified by PCR. The library was pooled and sequenced on a NovaSeq6000 (Illumina) instrument following the manufacturer's instructions. Sample qualification, library preparation, and sequencing were conducted by Macrogen Inc. (Korea).

After a quality control check for the raw data with FastQC (a quality control tool for high throughput sequence data) v0.11.19 (Babraham Bioinformatics), the adapter sequences, low-quality sequences (<Q20), and short sequences (<20 bp) were trimmed by Cutadapt v2.7 (Burrows-Wheeler Aligner). The remaining clean reads were aligned to the mouse mm10 (GCF_000000055.19, GRCm38.p6) reference genome using the aligner BWA MEM[34]. The SNPs and GTs of the variants were called separately for each sample by using the GATK v4.1.5.0 (Broad Institute)[35] tool HaplotypeCaller v4.1.0.0 and filtered with SelectVariants v4.0.8.0. The shared germline SNP indels were visualized by shinyCircos (RStudio)[36].

Chromosomal copy numbers were calculated using a method described previously[37]. Briefly, read counts were calculated using stools, and each chromosome read count was divided by the autosomes' average read count of the sample. Then, the chromosome value of each sample was normalized with the reference male sample.

**Sanger sequencing**. To detect SNPs between FVB/N and C57BL/6 on chromosome 2 and X chromosome, PCR reactions were performed under the following conditions: one cycle at 98 °C for 30 s, then 35 cycles at 98 °C for 10 s, 58 °C for 10 s, 72 °C for 2 min, and followed by one cycle at 72 °C for 5 min. The chromosome 2 primer set was the same set used by Miseq. For the X chromosome, the following primers were used: mX F-AAGGGTGATGGATATACGCC and mX R-CACAGAGGCACAGAAACAAC. The PCR products were purified, sequenced, and analyzed by Sequencher v5.0 (GeneCodes).

**RNA sequencing and data analysis**. RNAs were extracted using the RNeasy Mini kit (Qiagen) and qualified RNAs were used as input for the Illumina TruSeq Stranded messenger RNA LT Sample Prep Kit (Illumina) and sequencing libraries were created according to the manufacturer's protocol. Briefly, the sequencing library is prepared by random fragmentation of synthesized cDNA. Adapter-ligated fragments are amplified with PCR. The library was pooled and sequenced on the NovaSeq6000 (Illumina) instrument under the manufacturer's instructions. Sample qualification, library preparation, and sequencing were conducted by Macrogen Inc. (Korea).

To analyze RNA-Seq data, the quality of the paired and raw data was evaluated with FastQC v0.11.19, and low-quality sequences (<20%) and short sequences (<20 kb) were removed by using Cutadapt v2.7. The remained clean reads were mapped to the C57BL/6 mouse genome (PRJNA20689) using the aligner STAR v2.7.3a[38], and gene expressions were measured with Rsubread's feature count v3.6[39]. Furthermore, a Bioconductor package, Deseq2 v1.26.0, was used to analyze the differentially expressed genes according to default criteria[40]. The common genes of control and experimental groups which show similar expression patterns (fold change ≥2) have been selected, and the functional enrichment analysis of these differentially expressed genes was performed using the Gene Ontology database[41,42]. Pathway annotation was performed by Panther pathway analysis[43].

**Embryo transfer**. Blastocysts were transferred into the uteri of pseudopregnant (E2.5) ICR or CD-1 females. Pregnancy was monitored by recipient weight 1 week after embryo transfer. Pregnancy was estimated when recipients gained ≥2 g. The recipients were monitored for successful implantation and euthanized. The implanted embryos were collected for further analysis.

**aCGH**. DNA was extracted using the Gentra Puregene tissue kit (Qiagen). Samples were measured for copy numbers using the Agilent SurerPrint array kit according to the manufacturer's instructions. The samples were hybridized to a microarray printed with oligonucleotide probes and examined by a SureScan microarray scanner. The array data were analyzed by the CytoGenomics Software (Alilgent). Sample qualification, library preparation, and sequencing were conducted by BioCore (Korea).

**CNV analysis in the independent laboratory**. A genomic library was established using the tail tips of the parental B6D2F1 mice. WGA was performed using the SurePlex DNA Amplification kit (Illumina). The next-generation sequencing algorithm was used to determine the CNV of both the control and experimental embryos. CNV analysis was performed using VeriSeq PGS Kit (Illumina) and MiSeq Reagent Kit v3 - PGS (Illumina). The total CNV for all 20 chromosomes was plotted to visualize the embryo ploidy. For control embryos, the whole blastocyst was used. For experimental embryos, either the entire embryo or a biopsy of 4–8 trophectoderm cells was used. WGA was performed at the Center for Reproductive Medicine Preimplantation Genetic Testing Laboratory of Weill Cornell Medicine. Weill Cornell Medicine's Applied Bioinformatics Core Laboratory Facility performed all bioinformatic analyses and sequencing.

**Statistics and reproducibility**. Data are presented as mean ± standard error of the mean (s.e.m.) or standard deviation (s.d.). Independent-group $t$ tests or Fisher's exact test for two groups or ANOVA with Tukey analysis for multiple comparisons were used in this study. All statistical analyses were performed using the GraphPad Prism software (version 5.02), with $P < 0.05$ or 0.01 considered significant.

**Reporting summary**. Further information on research design is available in the Nature Research Reporting Summary linked to this article.

## Data availability

Data supporting the findings of the current study are provided in the paper and Supplementary Data 2 and 3 or have been deposited in the Sequence Read Archive (SRA) at NCBI with the accession numbers PRJNA787029, PRJNA787035, and PRJNA787047 for RNA-seq, PRJNA792598 and PRJNA793719 for Miseq, and PRJNA793729, PRJNA793731, PRJNA793733, PRJNA793736, PRJNA793738, PRJNA793739, PRJNA793741, and PRJNA793742 for WES. All source data underlying the graphs and charts shown in the main figures are presented in Supplementary Data 1. The uncropped gel images for Supplementary Figs. 6a, b and 8c are included in the Supplementary Information file as Supplementary Figs. 10–12.

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

# ARTICLE

31. So, S., Karagozlu, M. Z., Lee, Y. & Kang, E. Fasudil increases the establishment of somatic cell nuclear transfer embryonic stem cells in mouse. *J. Anim. Reprod. Biotechnol.* **35**, 21–27 (2020).

32. Ma, H. et al. Correction of a pathogenic gene mutation in human embryos. *Nature* **548**, 413–419 (2017).

33. Kang, E. et al. Age-related accumulation of somatic mitochondrial DNA mutations in adult-derived human iPSCs. *Cell Stem Cell* **18**, 625–636 (2016).

34. Li, H. & Durbin, R. Fast and accurate short read alignment with Burrows-Wheeler transform. *Bioinformatics* **25**, 1754–1760 (2009).

35. McKenna, A. et al. The Genome Analysis Toolkit: a MapReduce framework for analyzing next-generation DNA sequencing data. *Genome Res.* **20**, 1297–1303 (2010).

36. Yu, Y., Yao, W., Wang, Y. & Huang, F. shinyChromosome: an R/Shiny application for interactive creation of non-circular plots of whole genomes. *Genomics Proteomics Bioinformatics* **17**, 535–539 (2019).

37. Treff, N. R. et al. Next generation sequencing-based comprehensive chromosome screening in mouse polar bodies, oocytes, and embryos. *Biol. Reprod.* **94**, 76 (2016).

38. Dobin, A. et al. STAR: ultrafast universal RNA-seq aligner. *Bioinformatics* **29**, 15–21 (2013).

39. Liao, Y., Smyth, G. K. & Shi, W. featureCounts: an efficient general purpose program for assigning sequence reads to genomic features. *Bioinformatics* **30**, 923–930 (2014).

40. Love, M. I., Huber, W. & Anders, S. Moderated estimation of fold change and dispersion for RNA-seq data with DESeq2. *Genome Biol.* **15**, 550–558 (2014).

41. Ashburner, M. et al. Gene ontology: tool for the unification of biology. The Gene Ontology Consortium. *Nat. Genet.* **25**, 25–29 (2000).

42. The Gene Ontology Consortium. The Gene Ontology Resource: 20 years and still GOing strong. *Nucleic Acids Res.* **47**, D330–D338 (2019).

43. Mi, H., Muruganujan, A., Ebert, D., Huang, X. & Thomas, P. D. PANTHER version 14: more genomes, a new PANTHER GO-slim and improvements in enrichment analysis tools. *Nucleic Acids Res.* **47**, D419–D426 (2019).

## Acknowledgements

We thank Deokhoon Kim and Mustafa Zafer Karagozlu for providing technical support. Research in E.K.'s laboratory was supported by grants from the National Research Foundation of Korea (grant no. NRF-2018R1A2B3001244) and intramural grants from the Asan Institute for Life Sciences, Asan Medical Center (grant nos. 2019IP755 and 2020IP0021). The study in S.M.'s laboratory was supported by grants from the Burroughs Wellcome Fund and the National Institutes of Health (grant no. RO1AG062459) as well as by the institutional funds from the Oregon Health and Science University.

## Author contributions

Conceptualization, E.K., Y. Lee, G.D.P., and S.M.; investigation, E.K., Y. Lee, S, K., B.T., J.C., S.S., J.H., A.T., P.X., N.M.-G., J.D.Y., D.L., A.M., R.T.-H., H.M., C.V.D., H.D., Y. Li, J.X., F.X., A.K., P.A., and S.M.; writing—original draft, E.K., Y. Lee, S.M., and A.T.; writing—review and editing, E.K., S.M., and Y. Lee; supervision, E.K., S.M., and G.D.P.; funding acquisition, E.K. and S.M.

## Competing interests

The authors declare no competing interests.
