## [Peer Review File · Communications Biology]

Reviewers' comments:

Reviewer #1 (Remarks to the Author):

The authors studied different aspects of somatic cell haploidization in enucleated oocytes, using the mouse model, in search for the optimal method to support nuclear haploidization and reprogramming. They demonstrate that the replacement of meiotic spindles in MII oocytes with nuclei of somatic cells in the G1 phase can result in the formation of de novo spindles and haploidization. When fertilized in vitro, some of these reconstructed oocytes can form normally appearing zygotes, develop to the blastocyst stage and generate embryonic stem cells. Live offspring resulted from the transfer of these blastocyst to recipient mice, although the efficacy was low. These observations offer an alternative strategy for generating functional oocytes from somatic cells.

The subject of the study is topical, and the manuscript is well written. The conclusions drawn are clear, and the literature is nearly complete. I would just suggest to discuss the recently published data showing the existing differences, concerning the fidelity of chromatid separation and nuclear reprogramming in reconstructed oocytes, between mice and humans (Reproduction & Fertility 2021; 1: H1-H8. doi: 10.1530/RAF-20-0039).

After the incorporation of this minor change, the manuscript can be accepted for publication in its present form.

Reviewer #2 (Remarks to the Author):

The manuscript of Lee and colleagues investigated the possibility of producing embryos from SCNT reconstructed eggs undergoing a reductional division coupled with IVF. Unfortunately, I found the manuscript rather disappointing. In my opinion, the issue are not the results themselves but their presentation. Overall, the results are not presented in a very clear way, the rationale for the experimental design is not always explained and sometimes I found the text slightly misleading. I am also disappointed that the work of Tateno and colleagues is not mentioned and discussed (Human Reproduction, Volume 18, Issue 3, March 2003, Pages 472–473). It would be particularly interesting to see how the results reported here correlate with Tateno's prediction and model, if at all. The fact that not all donor genotypes produced pups indicate that apart from the reductional division, which is the main topic of the manuscript, other factors are at play. However, because the BDF1 genotype was not used consistently throughout the analyses, it is hard to draw a definite conclusion on both the segregation and the reprogramming. The actual numbers of embryos analysed/used are sometimes not reported in the text. Finally, I would also appreciate if some summarizing table with efficiencies was given (a percentage of a percentage gives the impression of better results than a percentage of a total). I would also recommend the authors to re-check the statistics and include the exact test used for individual experiments as my calculations do not always support the author's conclusion on the statistical significance. However, I could be wrong. Next, I would like to see a much stronger discussion. As it stands, it is rather a summary than a discussion. What would be the mechanism of a non-random chromosome segregation? What are the results obtained by others and how do the reported results fit? The reductional division induced by oocyte is an interesting topic.

Below are some specific comments. Nevertheless, I would recommend the authors to recheck the whole manuscript for the accuracy of information.

L. 38-39 The sentence is not very accurate. The MS does not really describe the production of functional oocytes. The eggs were produced in vivo and then reconstructed with a somatic cell in vitro. Please rephrase this sentence.

L41-45 I believe what the authors are describing applies to an asymmetric division of oocytes only. It is not a general rule to all types of a reductional division and therefore the paragraph should be rephrased.

L46-49 The authors need to include a citation.

L51-53 A more detailed, or specific, description of the experimental scheme should be included. It is not clear what the actual experiment was.

L83-85 There is no direct evidence for somatic chromosome pairing. The images are not really

supporting this claim. Moreover, the statement seems rather misleading in the context of the text that follows: on L117-119 the authors describe that only 44% of reconstructed eggs form a spindle.

L99-100 If the authors mention modifying the SCNT protocol, what was the initial method? A direct injection?

L101-103 What was the rationale for using these compounds?

L121-126 The actual numbers are not given. What would be the rationale behind a change of gene expression at the stages treated?

L130-132 What treatments were used? The statement is not very clear. The numbers are not given.

L150-168 This part is difficult to understand.

L153-154 How exactly does this indicate haploidy?

L155-156 I am not sure what the authors wanted to say.

L162-164 What exactly does the "maternal genome" refer to in this context?

L228 – What is the efficiency of ESC line derivation?

L384 C57BL/6 is inaccurate as Charles River sells both N and J strains, which differ significantly. Likewise, were BDF1 mice CrI or J? How old were the animals? Since not all genetic backgrounds gave rise to the pups, this information might be vital for other authors.

Reviewer #1

Referee comment	The authors studied different aspects of somatic cell haploidization in enucleated oocytes, using the mouse model, in search for the optimal method to support nuclear haploidization and reprogramming. They demonstrate that the replacement of meiotic spindles in MII oocytes with nuclei of somatic cells in the G1 phase can result in the formation of de novo spindles and haploidization. When fertilized in vitro, some of these reconstructed oocytes can form normally appearing zygotes, develop to the blastocyst stage and generate embryonic stem cells. Live offspring resulted from the transfer of these blastocyst to recipient mice, although the efficacy was low. These observations offer an alternative strategy for generating functional oocytes from somatic cells. The subject of the study is topical, and the manuscript is well written. The conclusions drawn are clear, and the literature is nearly complete. I would just suggest to discuss the recently published data showing the existing differences, concerning the fidelity of chromatid separation and nuclear reprogramming in reconstructed oocytes, between mice and humans (Reproduction & Fertility 2021; 1: H1-H8. doi: 10.1530/RAF-20-0039). After the incorporation of this minor change, the manuscript can be accepted for publication in its present form.
Response	Thank you for the reviewer's suggestion. We added the discussion including the article that you suggested, and others related to somatic haploidy and reprogramming in the revised manuscript (pages 16-17, lines 343-368). See also below. “The somatic chromosome segregation and nuclear remodeling/reprogramming could be crucial for the successful generation of offspring harboring haploid genomes derived from somatic cells. After the diploid somatic nucleus could be segregated to haploid in ooplasm, then the proper nuclear remodeling/reprogramming is required, resulting in the full development of SH embryo and the generation of SH offspring. Normal nuclear reprogramming to produce cloning animals has been proven in multiple species, even the efficiency was low¹⁹. However, there are debates for the success of chromosome segregation of somatic nucleus^{2, 20}. The first attempt of somatic haploidization was performed using the mature oocytes (MII) in human⁷. Reconstructed oocytes were fertilized, resulting in the extrusion of PPB. These PPBs were confirmed a single fluorescence signal by fluorescence in-situ hybridization in five chromosomes, which could indicate the segregation of homologous chromosomes. Other investigators tried somatic haploidization using the immature oocytes (germinal vesicle, GV) in humans and mice³. After transplantation of somatic cells into the GV ooplasm, the extrusion of the first polar body was observed in both humans and mice. However, the rate of polar body extrusion was under 1% and abortive metaphase plates were observed in mice⁴. Later, somatic haploidy was tried using MII oocytes in mice by Tateno and colleagues^{6, 20}. Reconstructed oocytes were chemically activated and most oocytes, that extruded PPB, failed to leave a haploid number of chromosomes. None of the reconstructed chromosomes showed metaphase-like-array before PPB extrusion. Based on the results, they claimed there was no opportunity for the pairing of somatic homologous chromosomes in MII oocytes and non-random chromosome segregation could not have happened in MII oocytes²⁰. Our study also used MII oocytes and demonstrated that reconstructed oocytes showed a metaphase-like spindle–chromosomal complex and the PPB was confirmed as haploidy by copy number analysis. Proper segregation of homologous chromosomes was observed in all analyzed SH zygotes, the number of the properly segregated chromosome in each SH embryos was 9 to 20. We

suggested that our modified SCNT protocol could assist the proper chromosome segregation.”

We also added discussion about the low developmental efficiency of SH embryos regarding reprogramming. We supposed that the mismatch of the reprogramming cycle between sperm and somatic haploidy genome could induce the developmental arrest in SH embryos.

For efficient presentation, the embryo development result of regular SCNT in Supplementary Fig. 9a in original manuscript was moved to Supplementary Table 1 in revised manuscript. We added a detailed explanation in the result (pages 7-8, lines 149-153) and the discussion part (pages 18-19, lines 399-408). Please see also below.

Supplementary Table. 1. Development of preimplantation embryos with various chemical treatments.

Group	Resting time after SCNT			Treatment during IVF	Treatment during overnight	Oocytes N	Fertilized N (%)	2PN formation N (%)	2PN/1PPB formation N (%)	2-cells N (%)	Morula N (%)	Blastocysts N (%)	
	Total	RA treatment time	RA RS-1 treatment										
Intact-IVF						227	218 (96)	210 (96)	206 (94)	199 (97)	152 (76)	138 (91)	
Regular SCNT						60	-	39 (65)	-	32 (82)	16 (50)	11 (69)	
	0.5 h					117	116 (99)	36 (31)	0 (0)	-	-	-	
	1 h					81	78 (96)	29 (37)	3 (4)	-	-	-	
	1.5 h					91	88 (97)	56 (64)	11 (13)	-	-	-	
	2 h					95	87 (92)	66 (76)	16 (19) ^a	-	-	-	
	3 h					87	86 (99)	39 (45)	8 (9)	-	-	-	
						198	184 (93)	144 (78)	31 (17)	29 (94)	14 (48)	4 (29)	
				Sc	Sc	111	109 (98)	79 (72)	24 (22)	23 (96)	10 (43)	4 (40)	
				Sc+Fa	Sc+Fa	187	184 (98)	142 (77)	58 (32) ^b	54 (93)	26 (48)	10 (38)	
				Sc+Fa	Sc+Fa	112	108 (96)	79 (73)	35 (32)	-	-	-	
				Subtotal	-	312	305 (98)	230 (75)	103 (34)	-	-	-	
NT-IVF													
	2 h			RA	0.5 h	Sc+Fa	Sc+Fa	92	89 (97)	63 (71)	46 (52) ^c	-	-
				RA	1 h	Sc+Fa	Sc+Fa	94	83 (88)	53 (64)	20 (24)	-	-
				RA	2 h	Sc+Fa	Sc+Fa	78	70 (90)	45 (64)	17 (24)	-	-
				RA	0.5 h	Sc+Fa	Sc+Fa	208	202 (97)	144 (71)	104 (51) ^d	95 (91)	40 (42)
						RS-1	Sc+Fa	Sc+Fa	155	153 (99)	117 (76)	62 (40)	57 (92)
				RA	0.5 h	RS-1	Sc+Fa	Sc+Fa	158	155 (98)	123 (79)	78 (50)	73 (94)
				RA	0.5 h		Sc+Fa+RS-1	Sc+Fa	124	118 (95)	102 (87)	82 (69) ^e	74 (90)
				RA	0.5 h		Sc+Fa+RS-1	Sc+Fa+RS-1	229	221 (97)	180 (81)	148 (67) ^e	135 (91)
				RA	0.5 h	RS-1	Sc+Fa+RS-1	Sc+Fa+RS-1	188	183 (97)	124 (68)	72 (39)	68 (94)
							Sc+Fa+RS-1	Sc+Fa+RS-1	229	221 (97)	180 (81)	148 (67) ^e	135 (91)
				RA	0.5 h	RS-1	Sc+Fa+RS-1	Sc+Fa+RS-1	188	183 (97)	124 (68)	72 (39)	68 (94)

“The blastocyst rates were 91% (n = 138/152; blastocysts/morula) and 69% (n = 11/16) in intact IVF and regular SCNT, respectively. As we expected, intact IVF showed a significantly higher blastocyst rate than that of aNT-IVF (Fig. 2h and Supplementary Table 1). However, the rate of blastocysts tended to be higher in regular SCNT (69%) compared to aNT-IVF (50%).”

“The third explanation is the mismatch of the reprogramming cycle between sperm and somatic haploidy genome. Initially, we expected that the reprogramming of NT-IVF might be better than regular SCNT because one nucleus of the reconstructed embryo was originated from a germ cell (sperm). However, the blastocyst development from the morula was ~50% in NT-IVF, which was tended to be lower than regular SCNT (69%). Therefore, we supposed that if the embryo nuclei were originated from the same cell type, such as oocyte and sperm nucleus in intact IVF embryos or somatic cell nuclei in regular SCNT embryos, the reprogramming and embryo development could be more effective. Even somatic haploidization was successful in NT-IVF embryos, sperm and somatic genome harbored different nuclear statuses for reprogramming, which could make the development arrest.”

	somatic cells: past, present and future. Reproduction and Fertility 2, H1-H8 (2021). 3. Palermo, G. D., Takeuchi, T. & Rosenwaks, Z. Oocyte-induced haploidization. Reprod. Biomed. Online 4, 237-242 (2002). 4. Fulka, J., Martinez, F., Tepla, O., Mrazek, M. & Tesarik, J. Somatic and embryonic cell nucleus transfer into intact and enucleated immature mouse oocytes. Hum. Reprod. 17, 2160-2164 (2002). 6. Tateno, H., Akutsu, H., Kamiguchi, Y., Latham, K. E. & Yanagimachi, R. Inability of mature oocytes to create functional haploid genomes from somatic cell nuclei. Fertil. Steril. 79, 216-218 (2003). 7. Tesarik, J., Nagy, Z. P., Sousa, M., Mendoza, C. & Abdelmassih, R. Fertilizable oocytes reconstructed from patient's somatic cell nuclei and donor ooplasts. Reprod. Biomed. Online 2, 160-164 (2001). 19. Matoba, S. & Zhang, Y. Somatic Cell Nuclear Transfer Reprogramming: Mechanisms and Applications. Cell. Stem Cell. 23, 471-485 (2018). 20. Tateno, H., Latham, K. E. & Yanagimachi, R. Reproductive semi-cloning respecting biparental origin. A biologically unsound principle. Hum. Reprod. 18, 472-473 (2003).
--	--

Reviewer #2

Comment No. 1	
Referee comment	The manuscript of Lee and colleagues investigated the possibility of producing embryos from SCNT reconstructed eggs undergoing a reductional division coupled with IVF. Unfortunately, I found the manuscript rather disappointing. In my opinion, the issue are not the results themselves but their presentation. Overall, the results are not presented in a very clear way, the rationale for the experimental design is not always explained and sometimes I found the text slightly misleading. I am also disappointed that the work of Tateno and colleagues is not mentioned and discussed (Human Reproduction, Volume 18, Issue 3, March 2003, Pages 472–473). It would be particularly interesting to see how the results reported here correlate with Tateno’s prediction and model, if at all.
Response	First, we apologize for the improper presentation. The entire manuscript has been revised to improve the presentation. We added more discussion regarding the arguments of Tateno and colleagues’ articles and others related to somatic chromosomes haploidy and nuclear reprogramming. Tateno’s and our studies used MII oocytes for somatic haploidization in mice. Tateno’s team claimed the limitations of chromosomal segregation and semi-cloning (the retaining chromosomes of just one parental origin from the somatic genome) during somatic haploidization using MII oocytes. Based on their report, the chance of semi-cloning was rare, therefore semi-cloning was hard to happen in the cloning with somatic haploidization. While we demonstrated a metaphase-like spindle–chromosomal complex in reconstructed oocytes and the proper chromosomal segregation after somatic haploidization. Further, we did not expect the semi-cloning after haploidization, because we assumed that it was not necessary to transmit only just one parental origin after somatic haploidization. We focused on the segregation of somatic homologous chromosomes to PPB and embryo, resulting in that an average of 76%

of the homologous chromosome is properly segregated. We added this discussion on pages 16-17, lines 343-378. Please see also below.

“The somatic chromosome segregation and nuclear remodeling/reprogramming could be crucial for the successful generation of offspring harboring haploid genomes derived from somatic cells. After the diploid somatic nucleus could be segregated to haploid in ooplasm, then the proper nuclear remodeling/reprogramming is required, resulting in the full development of SH embryo and the generation of SH offspring. Normal nuclear reprogramming to produce cloning animals has been proven in multiple species, even the efficiency was low¹⁹.

However, there are debates for the success of chromosome segregation of somatic nucleus²²⁰. The first attempt of somatic haploidization was performed using the mature oocytes (MII) in human⁷. Reconstructed oocytes were fertilized, resulting in the extrusion of PPB. These PPBs were confirmed a single fluorescence signal by fluorescence in-situ hybridization in five chromosomes, which could indicate the segregation of homologous chromosomes. Other investigators tried somatic haploidization using the immature oocytes (germinal vesicle, GV) in humans and mice³. After transplantation of somatic cells into the GV ooplasm, the extrusion of the first polar body was observed in both humans and mice. However, the rate of polar body extrusion was under 1% and abortive metaphase plates were observed in mice⁴. Later, somatic haploidy was tried using MII oocytes in mice by Tateno and colleagues^{6,20}. Reconstructed oocytes were chemically activated and most oocytes, that extruded PPB, failed to leave a haploid number of chromosomes. None of the reconstructed chromosomes showed metaphase-like-array before PPB extrusion. Based on the results, they claimed there was no opportunity for the pairing of somatic homologous chromosomes in MII oocytes and non-random chromosome segregation could not have happened in MII oocytes²⁰. Our study also used MII oocytes and demonstrated that reconstructed oocytes showed a metaphase-like spindle–chromosomal complex and the PPB was confirmed as haploidy by copy number analysis. Proper segregation of homologous chromosomes was observed in all analyzed SH zygotes, the number of the properly segregated chromosome in each SH embryos was 9 to 20. We suggested that our modified SCNT protocol could assist the proper chromosome segregation.

Further, Tateno and colleagues suggested that the chance of retaining chromosomes of just one parental origin, known as semi-cloning, is rare, which could be less than 1×2^{-20} for 20 chromosome pairs in mice, therefore, semi-cloning did not have any advantage for the cloning with somatic haploidization²⁰. Therefore, we did not concern about the semi-cloning during haploidization because we assumed that it was not necessary to transmit only just one parental origin after somatic haploidization. We demonstrated that the contribution of somatic chromosomes in SH embryo was random between maternal or paternal alleles in the somatic genome. Instead of the odds of semi-cloning, we focused on the proper segregation of somatic homologous chromosomes to PPB and embryo, resulting in that an average of 76% of the homologous chromosome was properly segregated in 15 SH embryos (45-100% of range in each embryo).”

2. Tesarik, J., Mendoza, C. & Mendoza-Tesarik, R. Human artificial oocytes from patients' somatic cells: past, present and future. *Reproduction and Fertility* 2, H1-H8 (2021).
3. Palermo, G. D., Takeuchi, T. & Rosenwaks, Z. Oocyte-induced haploidization. *Reprod. Biomed. Online* 4, 237-242 (2002).
4. Fulka, J., Martinez, F., Tepla, O., Mrazek, M. & Tesarik, J. Somatic and embryonic cell nucleus transfer into intact and enucleated immature mouse oocytes. *Hum. Reprod.* 17, 2160-2164 (2002).

	6. Tateno, H., Akutsu, H., Kamiguchi, Y., Latham, K. E. & Yanagimachi, R. Inability of mature oocytes to create functional haploid genomes from somatic cell nuclei. Fertil. Steril. 79, 216-218 (2003). 7. Tesarik, J., Nagy, Z. P., Sousa, M., Mendoza, C. & Abdelmassih, R. Fertilizable oocytes reconstructed from patient's somatic cell nuclei and donor ooplasts. Reprod. Biomed. Online 2, 160-164 (2001). 19. Matoba, S. & Zhang, Y. Somatic Cell Nuclear Transfer Reprogramming: Mechanisms and Applications. Cell. Stem Cell. 23, 471-485 (2018). 20. Tateno, H., Latham, K. E. & Yanagimachi, R. Reproductive semi-cloning respecting biparental origin. A biologically unsound principle. Hum. Reprod. 18, 472-473 (2003).
Comment No. 2	
Referee comment	The act that not all donor genotypes produced pups indicate that apart from the reductional division, which is the main topic of the manuscript, other factors are at play. However, because the BDF1 genotype was not used consistently throughout the analyses, it is hard to draw a definite conclusion on both the segregation and the reprogramming.
Response	We agreed with the reviewer's opinion. We tested several combinations of somatic cells and sperms with different species, then the only combination with BDF1 somatic cells and BDF1 sperm could generate the SH-pups. We suggested that species was important for producing full-term births experimentally. However, BDF1 was not usually used for the genetic analyses, therefore, it is hard to confirm the haploidization genetically in SH pups, which is a limitation of our study. We added this discussion on page 19, lines 409-416. Please see also below. “The generation of SH offspring was available by only the combination of BDF1 somatic cells and BDF1 sperm in this study. Initially, we expected that the combination of B6/FVB somatic cells and BDF1 sperm could produce the SH offspring because hybrid species are efficient to produce offspring experimentally than inbred species²⁴. However, this combination failed. Another combination, BDF1 somatic cells, and BDF1 sperm were successful, suggesting that the species was crucial for producing full-term births experimentally. Unfortunately, BDF1 was not used consistently throughout the genetic analyses, therefore, it is hard to confirm the haploidization in SH pups, which is a limitation of our study.” 24. Kirchain, S. M., Hayward, A. M., Mkandawire, J. M., Qi, P. & Burds, A. A. Comparison of tetraploid blastocyst microinjection of outbred Crl:CD1(ICR), hybrid B6D2F1/Tac, and inbred C57BL/6NTac embryos for generation of mice derived from embryonic stem cells. Comp. Med. 58, 145-150 (2008).
Comments No. 3	
Referee comment	The actual numbers of embryos analyzed/used are sometimes not reported in the text. Finally, I would also appreciate if some summarizing table with efficiencies was given (a percentage of a percentage gives the impression of better results than a percentage of a total). I would also recommend the authors to re-check the statistics and include the exact test used for individual experiments as my calculations do not always support the author's conclusion on the statistical significance. However, I could be wrong.

Response	We added the actual number of embryos in ‘Fasudil, retinoic acid, and RS-1 promote the segregation of homologous chromosomes’ section of the result part (pages 5-8, lines 98-153). Further, we made a new summarizing table for the efficiency of embryo development calculated by a percentage of a percentage as your suggestion. Please see supplementary table 2 in the supplementary information file. We also revised the result part (page 7, lines 147-149) and added the legend in the supplementary information file (page 15, lines 125-130). “When the blastocyst rate was calculated from MII oocytes, 12% oocytes (n = 27/229) could develop blastocysts with the aNT-IVF protocol, while only 4% (n = 4/198) with the non-treated NT-IVF one (Supplementary Table 2).” “Supplementary Table 2. The efficiency of embryo development with various chemical treatments.    Group Resting time after SCNT Treatment during IVF Treatment during overnight Oocytes N Fertilization % 2PN/1PPB formation % 2-cells % Morula % Blastocysts %   RA treatment RS-1 treatment     Intact-IVF     227 96 90 88 67 61       198 93 16 15 7 2   NT-IVF   Sc Sc 111 98 22 21 9 4     Sc+Fa Sc+Fa 187 98 31 29 14 5   RA  Sc+Fa Sc+Fa 208 97 49 45 19 9    RS-1 Sc+Fa Sc+Fa 155 99 40 36 15 4   RA RS-1 Sc+Fa Sc+Fa 158 98 49 46 17 5   RA  Sc+Fa+RS-1  124 95 66* 59 21 7   RA  Sc+Fa+RS-1 Sc+Fa+RS-1 229 97 65* 59 24 12#   RA RS-1 Sc+Fa+RS-1 Sc+Fa+RS-1 188 97 38 36 12 6    The efficiency was calculated by a percentage of a percentage. * or # indicate significantly higher 2PN/1PPB or blastocyst rate compared to non-treatment groups (P < 0.05, by Independent-group t-test).” We re-check the statistics and stated the exact test for individual experiments in the legend; Fig. 1e (page 36, line 781), 2b-e (page 38, lines 790-796), 2h (page 38, line 808), and 8a (page 48, line 883) in revised manuscript, and Supplementary Fig. 1b-c (page 1, lines 6 and 9), 8a (page 10, line 74), and 9d (page 13, line 92), Supplementary table 1 (pages 14-15, lines 112-123) and 2 (page 15, lines 129-130) in supplementary information file.	Group	Resting time after SCNT		Treatment during IVF	Treatment during overnight	Oocytes N	Fertilization %	2PN/1PPB formation %	2-cells %	Morula %	Blastocysts %	RA treatment	RS-1 treatment	Intact-IVF					227	96	90	88	67	61					198	93	16	15	7	2	NT-IVF			Sc	Sc	111	98	22	21	9	4			Sc+Fa	Sc+Fa	187	98	31	29	14	5	RA		Sc+Fa	Sc+Fa	208	97	49	45	19	9		RS-1	Sc+Fa	Sc+Fa	155	99	40	36	15	4	RA	RS-1	Sc+Fa	Sc+Fa	158	98	49	46	17	5	RA		Sc+Fa+RS-1		124	95	66*	59	21	7	RA		Sc+Fa+RS-1	Sc+Fa+RS-1	229	97	65*	59	24	12#	RA	RS-1	Sc+Fa+RS-1	Sc+Fa+RS-1	188	97	38	36	12	6
Group	Resting time after SCNT		Treatment during IVF	Treatment during overnight									Oocytes N	Fertilization %		2PN/1PPB formation %	2-cells %	Morula %	Blastocysts %																																																																																																	
	RA treatment	RS-1 treatment																																																																																																																		
Intact-IVF					227	96	90	88	67	61																																																																																																										
					198	93	16	15	7	2																																																																																																										
NT-IVF			Sc	Sc	111	98	22	21	9	4																																																																																																										
			Sc+Fa	Sc+Fa	187	98	31	29	14	5																																																																																																										
	RA		Sc+Fa	Sc+Fa	208	97	49	45	19	9																																																																																																										
		RS-1	Sc+Fa	Sc+Fa	155	99	40	36	15	4																																																																																																										
	RA	RS-1	Sc+Fa	Sc+Fa	158	98	49	46	17	5																																																																																																										
	RA		Sc+Fa+RS-1		124	95	66*	59	21	7																																																																																																										
RA		Sc+Fa+RS-1	Sc+Fa+RS-1	229	97	65*	59	24	12#																																																																																																											
RA	RS-1	Sc+Fa+RS-1	Sc+Fa+RS-1	188	97	38	36	12	6																																																																																																											
Comment No. 4																																																																																																																				
Referee comment	Next, I would like to see a much stronger discussion. As it stands, it is rather a summary than a discussion. What would be the mechanism of a non-random chromosome segregation? What are the results obtained by others and how do the reported results fit?																																																																																																																			
Response	We added discussion about somatic chromosome segregation compared to Tateno and colleagues’ articles, which claimed random chromosomal segregation of somatic genome in MII oocytes. Unlike Tateno’s article, our study demonstrated																																																																																																																			

	that somatic homologous chromosome were properly segregated to PPB and embryo reciprocally. We suggested our modified SCNT protocol could assist to induce the proper chromosome segregation. We added this discussion in pages 16-17, lines 357-378. Please see below. “Later, somatic haploidy was tried using MII oocytes in mice by Tateno and colleagues^{6,20}. Reconstructed oocytes were chemically activated and most oocytes, that extruded PPB, failed to leave a haploid number of chromosomes. None of the reconstructed chromosomes showed metaphase-like-array before PPB extrusion. Based on the results, they claimed there was no opportunity for the pairing of somatic homologous chromosomes in MII oocytes and non-random chromosome segregation could not have happened in MII oocytes²⁰. Our study also used MII oocytes and demonstrated that reconstructed oocytes showed a metaphase-like spindle–chromosomal complex and the PPB was confirmed as haploidy by copy number analysis. Proper segregation of homologous chromosomes was observed in all analyzed SH zygotes, the number of the properly segregated chromosome in each SH embryos was 9 to 20. We suggested that our modified SCNT protocol could assist the proper chromosome segregation. Further, Tateno and colleagues suggested that the chance of retaining chromosomes of just one parental origin, known as semi-cloning, is rare, which could be less than 1×2^{-20} for 20 chromosome pairs in mice, therefore, semi-cloning did not have any advantage for the cloning with somatic haplodization²⁰. Therefore, we did not concern about the semi-cloning during haplodization because we assumed that it was not necessary to transmit only just one parental origin after somatic haplodization. We demonstrated that the contribution of somatic chromosomes in SH embryo was random between maternal or paternal alleles in the somatic genome. Instead of the odds of semi-cloning, we focused on the proper segregation of somatic homologous chromosomes to PPB and embryo, resulting in that an average of 76% of the homologous chromosome was properly segregated in 15 SH embryos (45-100% of range in each embryo).”
Referee comment	The reductional division induced by oocyte is an interesting topic. Below are some specific comments. Nevertheless, I would recommend the authors to recheck the whole manuscript for the accuracy of information.
Response	We answered the specific comments below and rechecked the whole manuscript as your suggestion.
Comment No. 5	
Referee comment	L. 38-39 The sentence is not very accurate. The MS does not really describe the production of functional oocytes. The eggs were produced in vivo and then reconstructed with a somatic cell in vitro. Please rephrase this sentence.
Response	We apologize for the inaccuracy of the sentence. We edited it (page 1, lines 38-39). Please see also below. “Our finding may offer an alternative strategy for generating oocytes carrying somatic genomes.”

Comment No. 6	
Referee comment	L41-45 I believe what the authors are describing applies to an asymmetric division of oocytes only. It is not a general rule to all types of a reductional division and therefore the paragraph should be rephrased.
Response	We revised sentence to limit to oocytes only for this part (page 3, lines 43-45). Please see also below. “For generating meiosis II (MII) oocytes, the reductional MI division is followed by an equational MII division, similar to mitosis, where the sister chromatids are segregated to opposite spindle poles, eliminating one set into the second polar body (PB2).”
Comment No. 7	
Referee comment	L46-49 The authors need to include a citation.
Response	We apologize for to miss citation. We revised the manuscript and included the citations (page 3, lines 46-55). Please see also below. “However, the diploid somatic cells that transferred into enucleated oocytes to achieve the haploid nucleus resulted in limited development of the preimplantation embryos². In mice, the somatic cells such as cumulus or fibroblasts were transferred to immature germinal vesicle (GV) or mature MII oocytes to induce haploid chromosomes, but the reconstructed chromosomes exhibited abnormalities in the separation and alignment processes and embryo development was not observed³⁻⁶. In humans, the cumulus cells were injected into the MII ooplasm that was removed chromosomes of the oocyte. The reconstructed oocytes were fertilized with a spermatozoon resulting in the polar body extrusion, which showed the segregation of homologous chromosomes in several chromosomes. However, no further development was observed⁷.”  2. Tesarik, J., Mendoza, C. & Mendoza-Tesarik, R. Human artificial oocytes from patients' somatic cells: past, present and future. Reproduction and Fertility 2, H1-H8 (2021). 3. Palermo, G. D., Takeuchi, T. & Rosenwaks, Z. Oocyte-induced haploidization. Reprod. Biomed. Online 4, 237-242 (2002). 4. Fulka, J., Martinez, F., Tepla, O., Mrazek, M. & Tesarik, J. Somatic and embryonic cell nucleus transfer into intact and enucleated immature mouse oocytes. Hum. Reprod. 17, 2160-2164 (2002). 5. Chang, C. C., Nagy, Z. P., Abdelmassih, R., Yang, X. & Tian, X. C. Nuclear and microtubule dynamics of G2/M somatic nuclei during haploidization in germinal vesicle-stage mouse oocytes. Biol. Reprod. 70, 752-758 (2004). 6. Tateno, H., Akutsu, H., Kamiguchi, Y., Latham, K. E. & Yanagimachi, R. Inability of mature oocytes to create functional haploid genomes from somatic cell nuclei. Fertil. Steril. 79, 216-218 (2003). 7. Tesarik, J., Nagy, Z. P., Sousa, M., Mendoza, C. & Abdelmassih, R. Fertilizable oocytes reconstructed from patient's somatic cell nuclei and donor ooplasts. Reprod. Biomed. Online 2, 160-164 (2001).

Comment No. 8	
Referee comment	L51-53 A more detailed, or specific, description of the experimental scheme should be included. It is not clear what the actual experiment was.
Response	As your suggestion, we added detailed experimental information of the reference article. This article investigated somatic haploidization using mature oocytes (MII) in humans. Reconstructed oocytes were fertilized, and the PPB were extruded. The PPBs were confirmed the chromosome segregation in five chromosomes by fluorescence in situ hybridization (FISH), however, embryo development after the 2-cell stage was not observed in this paper. We added this information on page 3, lines 51-55. Please see also below. “In humans, the cumulus cells were injected into the MII ooplasm that was removed chromosomes of the oocyte. The reconstructed oocytes were fertilized with a spermatozoon resulting in the polar body extrusion, which showed the segregation of homologous chromosomes in several chromosomes. However, no further development was observed⁷.” Further, we also added the detail of our experimental scheme (pages 3-4, lines 56-63). Please see also below. “Here, we revisited the haploidization of diploid somatic chromosomes using the somatic cell nuclear transfer (SCNT) technique, in which the somatic cell nucleus was transferred into enucleated metaphase MII–arrested oocytes. We examined meiotic spindles in SCNT oocytes that were produced by transplanting a G₀/G₁ somatic cell depending on resting time after SCNT and confirmed the chromosome segregation after in vitro fertilization (NT-IVF). We also improved the rates of development of somatic haploid (SH) embryos with various combinations of chemicals or protein. Finally, the SH embryos were established embryonic stem cells (ESCs) and produced offspring.”
Comment No. 9	
Referee comment	L83-85 There is no direct evidence for somatic chromosome pairing. The images are not really supporting this claim. Moreover, the statement seems rather misleading in the context of the text that follows: on L117-119 the authors describe that only 44% of reconstructed eggs form a spindle.
Response	We apologize to make misleading. We revised the sentences (page 5, lines 86-87). Please see also below. “The chromosomal arrangement pattern in some SCNT spindles was similar to those observed in MI oocytes.” Further, we also revised the whole manuscript to remove the claim related to the chromosomal pairing. Figure 1a and 9 were also revised to eliminate the words related to chromosome pairing.

Comments No. 10	
Referee comment	L99-100 If the authors mention modifying the SCNT protocol, what was the initial method? A direct injection?
Response	We revised the manuscript to state the conventional SCNT protocol (page 5, lines 99-105). Please see also below. “The SCNT technique has been performed previously⁹. Briefly, a hemagglutinating virus of Japan envelope was applied to fuse the donor somatic cells with the enucleated oocytes. After resting time for 30 min to 1 h, the reconstructed oocytes were activated with strontium and HDAC inhibitors such as Trichostatin A or Scriptaid. Based on this conventional method, we modified the protocol for NT-IVF, which was extended resting time for 2 h. Additionally, the caffeine was treated during SCNT micromanipulation and resting time to prevent premature oocyte activation and to prompt spindle reformation in SCNT oocytes^{10, 11}.” 9. Kang, E. et al. Nuclear reprogramming by interphase cytoplasm of two-cell mouse embryos. Nature 509, 101-104 (2014). 10. Mitalipov, S. M. et al. Reprogramming following somatic cell nuclear transfer in primates is dependent upon nuclear remodeling. Hum. Reprod. 22, 2232-2242 (2007). 11. Tachibana, M. et al. Human embryonic stem cells derived by somatic cell nuclear transfer. Cell 153, 1228-1238 (2013).
Comment No. 11	
Referee comment	L101-103 What was the rationale for using these compounds?
Response	We treated fasudil, RA, and RS-1 to improve the somatic haploidization. We explained the rationale for use of each chemical with the related articles in the result part (page 6, lines 110-112; page 6, lines 118-120; page 7, lines 132-134). Please see also below. “The first, the fasudil was treated during IVF. Since ROCK supports spindle assembly in mature oocytes¹², fasudil might assist the spindle decomposition during fertilization and enhance the PPB extrusion through the regulation of microtubule polarity¹³.” 12. Duan, X. et al. Rho-GTPase effector ROCK phosphorylates cofilin in actin-mediated cytokinesis during mouse oocyte meiosis. Biol. Reprod. 90, 37 (2014). 13. Yu, C. H. et al. Measuring microtubule polarity in spindles with second-harmonic generation. Biophys. J. 106, 1578-1587 (2014). “RA initiates the entrance of the prophase of meiosis I during oogenesis^{14, 15}. Because we proposed that the premature chromosomes from the G0/G1 somatic cell could be similar to the prophase of meiosis I of the oocyte.” 14. Baltus, A. E. et al. In germ cells of mouse embryonic ovaries, the decision to enter meiosis precedes premeiotic DNA replication. Nat. Genet. 38, 1430-1434 (2006). 15. Nasiri, E., Mahmoudi, R., Bahadori, M. H. & Amiri, I. The Effect of Retinoic Acid on in vitro Maturation and Fertilization Rate of Mouse Germinal Vesicle Stage Oocytes. Cell. J. 13, 19-24 (2011). “Finally, we evaluated the effect of RS-1, which enhances the expression of the homologous

	recombinases, such as Rad51 and Dmc1¹⁶. The RS-1 could support the alignment of homologous chromosomes of the somatic nucleus in SCNT oocytes thus increasing PPB extrusion.” 16. Lan, W. H. et al. Rad51 facilitates filament assembly of meiosis-specific Dmc1 recombinase. Proc. Natl. Acad. Sci. U. S. A. 117, 11257-11264 (2020).
Comment No. 12	
Referee comment	L121-126 The actual numbers are not given. What would be the rationale behind a change of gene expression at the stages treated?
Response	We added the number of embryos (page 7, lines 137-138). Please see also below. “Treatment during IVF and overnight resulted in a significantly higher rate of 2PN/1PPB (67%, n = 148/221 vs. 51%, n = 104/202, $P < 0.05$; Fig. 2e).” Because RS-1 enhances the homologous recombination, we expected to increase PPB extrusion through the improved alignment of homologous chromosomes for somatic haploidization. Although RS-1 is known to increase the expression of Rad51 and Dmc1, we focused on the increased rate of PPB extrusion rather than the expression of genes when RS-1 was treated. We explained the rationale to use RS-1 in this study in the result part (page 7, lines 132-134). Please see also below. “Finally, we evaluated the effect of RS-1, which enhances the expression of the homologous recombinases, such as Rad51 and Dmc1¹⁶. The RS-1 could support the alignment of homologous chromosomes of the somatic nucleus in SCNT oocytes thus increasing PPB extrusion.”
Comment No. 13	
Referee comment	L130-132 What treatments were used? The statement is not very clear. The numbers are not given.
Response	We apologize for the unclear statement. We added the modified protocol and the number of embryos for Figure 2h (page 7, line 139-147). Please see also below. “Based on the results of these treatments, we established an advanced (a)NT-IVF protocol; RA was treated for 30 min after SCNT, rested SCNT oocytes for 2 h before IVF, and scriptaid, fasudil, and RS-1 were treated during IVF and the overnight culture (Fig. 2f). These treatments could promote the spindle reconstruction and formation of normal SH zygotes. The SH zygotes had a normal morphological development up to the blastocyst stage (Fig. 2g and Supplementary Video 1). The incorporation of these treatments significantly increased 2PN/1PPB (67%, n = 148/ 221, vs. 17%, n = 31/184; 2PN/1PPB/fertilized) and blastocyst (50%, n = 27 /54 vs. 29%, n = 4/14; blastocysts/morula) formation rates compared with non-treated ($P < 0.05$; Fig. 2h and Supplementary Table 1).”

Comment No. 14	
Referee comment	L150-168 This part is difficult to understand.
Response	We revised the whole sentences in this part (pages 8-9, lines 171-192). “The X chromosome was analyzed by Sanger sequencing (Supplementary Fig. 3a). The segregation of somatic FVB (red) and B6 (blue) homologous chromosomes from the somatic cells were observed in most PPBs and SH embryos (Fig. 3b). Initially, we hypothesized that FVB SNPs could only be detected in either PPB or SH embryos because the somatic donor was heterozygous. However, the exome sequencing showed the recombined homozygous SNPs in heterozygous somatic donors (Supplementary Fig. 3b), which could make to detect the FVB SNPs in both PPB and SH embryos. Therefore, if FVB SNPs were detected in SH embryos and their corresponding PPB, we also considered the proper segregation of somatic chromosomes. We first checked the zygosity of chromosomes in each PPB, resulting in that 10–20 chromosomes were homozygous, which could be haploidy segregated from somatic genomes (Fig.3c). Among them, either FVB or B6 chromosomes were identified randomly in each chromosome. Next, the number of properly segregated chromosomes into PPB and SH embryos was analyzed. In total, 9 to 20 homologous chromosomes were properly segregated between SH embryos and their corresponding PPBs (Fig.3d). Some chromosomes in PPBs showed heterozygosity (74%) or were not amplified (26%), suggesting that these homologous chromosomes were not separated and extruded to PPBs or remained in embryos (Fig. 3e). In the SH embryos, 66% and 68% of haploid chromosomes in the FVB/B6 and B6/FVB combinations were originated from the FVB strain respectively, suggesting that the somatic genome remaining in SH embryos after haploidization was more species-specific rather than maternally or paternally biased (Fig. 3f). We also analyzed the segregation for each chromosome in 15 SH embryos. Chromosome 1 was segregated properly in all 15 embryos, whereas the other 19 chromosomes were separated in 8–14 embryos (Supplementary Fig. 3c).”
Comment No. 15	
Referee comment	L153-154 How exactly does this indicate haploidy?
Response	We revised that sentence (page 9, lines 179-181). Please see also below. “We first checked the zygosity of chromosomes in each PPB, resulting in that 10–20 chromosomes were homozygous, which could be haploidy segregated from somatic genomes (Fig.3c).”
Comment No. 16	
Referee comment	L155-156 I am not sure what the authors wanted to say.

Response	We edited the sentences (pages 8-9, lines 174-179) and added a new figure as supplementary Fig.3b and the legend (page 4, lines 25-27 in the supplementary information file). Please see also below. “Initially, we hypothesized that FVB SNPs could only be detected in either PPB or SH embryos because the somatic donor was heterozygous. However, the exome sequencing showed the recombined homozygous SNPs in heterozygous somatic donors (Supplementary Fig. 3b), which could make to detect the FVB SNPs in both PPB and SH embryos. Therefore, if FVB SNPs were detected in SH embryos and their corresponding PPB, we also considered the proper segregation of somatic chromosomes.” “b, FVB SNPs detected in heterozygous (B6/FVB) somatic donor cell. Black or green bars indicate total FVB or homozygous SNPs, respectively.” b FVB SNPs in heterozygous (B6/FVB) somatic donor cell  Total FVB SNPs Homozygous FVB SNPs
Comment No. 17	
Referee comments	L162-164 What exactly does the “maternal genome” refer to in this context?
Responses	We edited the sentence (page 9, lines 187-190). Please see also below. “In the SH embryos, 66% and 68% of haploid chromosomes in the FVB/B6 and B6/FVB combinations were originated from the FVB strain respectively, suggesting that the somatic genome remaining in SH embryos after haploidization was more species-specific rather than maternally or paternally biased (Fig. 3f).”
Comment No. 18	
Referee comment	L228 What is the efficiency of ESC line derivation?
Response	We added the efficiency of ESC derivation in the result part (page 12, lines 255-257) as Supplementary Fig.8a. We also revised the figure legend in the supplementary information file (page 10, lines 72-74). Please see also below. “We generated several SH-ESC lines, and the efficiency of ESC derivation was 7% (4 ESCs/55 NT-IVF BL), significantly lower than that of ESC from IVF embryos (75%, 13 ESCs/23 IVF BL) (Supplementary Fig. 8a).” “a, The efficiency of SH-ESCs derivation. The efficiency was 7%, which was significantly lower than that of ESC from IVF embryos (57%) ($P < 0.05$, by Independent-group t-test).”

Comment No. 19

Referee comment *L384 C57BL/6 is inaccurate as Charles River sells both N and J strains, which differ significantly. Likewise, were BDF1 mice Crl or J? How old were the animals? Since not all genetic backgrounds gave rise to the pups, this information might be vital for other authors.*

Response We added the information in the method part (page 20, lines 426-429). Please see also below.

“B6D2F1/Crl (C57BL/6N^{Crl} female x DBA/2N^{Crl} male) female mice (8 ~ 9-week-old, Charles River) were used as oocyte recipients. FVB/N female and male (8 ~ 9-week-old, Taconic Biosciences), C57BL/6N female and male (8 ~ 9-week-old, Charles River), and BDF1/Crl female (8 ~ 9-week-old) were used for the generation of somatic donor cells for SCNT.”

REVIEWERS' COMMENTS:

Reviewer #1 (Remarks to the Author):

The authors have addressed adequately the points raised by the referees and revised the paper accordingly. I recommend acceptance of this revised manuscript in its present form.